# Larger models yield better results? Streamlined severity classification of ADHD-related concerns using BERT-based knowledge distillation

**Ahmed Akib Jawad Karim**[1]\*, **Kazi Hafiz Md. Asad**[2☯], **Md. Golam Rabiul Alam**[1☯]

**1** Computer Science and Engineering, BRAC University, Dhaka, Bangladesh, **2** Electrical and Computer Engineering, North South University, Dhaka, Bangladesh

☯ These authors contributed equally to this work.
\* akibjawaad@gmail.com

## Abstract

This work focuses on the efficiency of the knowledge distillation approach in generating a lightweight yet powerful BERT-based model for natural language processing (NLP) applications. After the model creation, we applied the resulting model, LastBERT, to a real-world task—classifying severity levels of Attention Deficit Hyperactivity Disorder (ADHD)-related concerns from social media text data. Referring to LastBERT, a customized student BERT model, we significantly lowered model parameters from 110 million BERT base to 29 million-resulting in a model approximately 73.64% smaller. On the General Language Understanding Evaluation (GLUE) benchmark, comprising paraphrase identification, sentiment analysis, and text classification, the student model maintained strong performance across many tasks despite this reduction. The model was also used on a real-world ADHD dataset with an accuracy of 85%, F1 score of 85%, precision of 85%, and recall of 85%. When compared to DistilBERT (66 million parameters) and ClinicalBERT (110 million parameters), LastBERT demonstrated comparable performance, with DistilBERT slightly outperforming it at 87%, and ClinicalBERT achieving 86% across the same metrics. These findings highlight the LastBERT model's capacity to classify degrees of ADHD severity properly, so it offers a useful tool for mental health professionals to assess and comprehend material produced by users on social networking platforms. The study emphasizes the possibilities of knowledge distillation to produce effective models fit for use in resource-limited conditions, hence advancing NLP and mental health diagnosis. Furthermore underlined by the considerable decrease in model size without appreciable performance loss is the lower computational resources needed for training and deployment, hence facilitating greater applicability. Especially using readily available computational tools like Google Colab and Kaggle Notebooks. This study shows the accessibility and usefulness of advanced NLP methods in pragmatic world applications.

**Data Availability Statement:** All relevant data underlying the results presented in this study are publicly available and can be accessed from the Kaggle repository at the following URL: https://

www.kaggle.com/datasets/akibjawad/adhd-related-concerns.

**Funding:** The author(s) received no specific funding for this work.

**Competing interests:** The authors have declared that no competing interests exist.

# 1 Introduction

NLP is flourishing right now; big language models (LLMs) are quite useful for addressing many language challenges. These models are costly and difficult to teach, though, since they are generally somewhat huge. To address this issue, our research aimed to produce a smaller BERT model that can perform comparably to larger LLMs. This model was developed using knowledge distillation, where a student model learns from a teacher model. In our study, the teacher model was BERT large (340M), and the student model was BERT base with a custom configuration (29M). This new student model, for simplicity, is called LastBERT. Following that, the distilled model was trained and tested on several benchmark datasets, including the General Language Understanding Evaluation (GLUE) benchmark, which fully evaluated the model's performance over many NLP tasks. These tests showed that our student model performed competitively relative to the instructor model and other common LLMs, therefore highlighting the success of our distillation method. The findings revealed that on sentiment analysis and text classification tasks, the LastBERT model may do really well.

Extending this, in a real-world setting we used the model to categorize degrees of ADHD-related concerns from social media posts. The neurodevelopmental disorder called ADHD is defined by symptoms of inattention, hyperactivity, and impulsivity. New research employing rich, natural language data in social media posts and other textual interactions has concentrated on using textual analysis to detect ADHD. Much research has shown how well NLP can detect ADHD symptoms, which helps to create scalable and effective diagnostic instruments. This use shows how well our distilled model conveys mental health problems using natural language processing. To assess our new student model, LastBERT, in this study we also ran two existing models, ClinicalBERT and DistilBERT. The results were respectable, indicating that our model can be an effective tool in NLP-based mental health diagnostics. This approach aligns with recent advancements in the field where NLP techniques have been successfully used to detect mental health conditions from social media data.

The primary contributions of this research are:

1. Developed *LastBERT*, a custom, lightweight BERT-based student model through knowledge distillation. LastBERT delivers high performance on par with larger models like BERT-large and BERT-base, while also achieving comparable results with smaller models such as DistilBERT and TinyBERT, despite having only 29 million parameters.

2. Rigorously evaluated LastBERT on six GLUE benchmark datasets to confirm its adaptability and robustness across diverse NLP tasks such as text classification, sentiment analysis, and paraphrase identification.

3. Created a specialized ADHD-related concerns dataset from the Reddit Mental Health dataset and applied LastBERT model to perform severity classification-—demonstrating the model's potential in addressing real-world challenges in mental health diagnostics.

4. Conducted a detailed comparative analysis with DistilBERT and ClinicalBERT on the ADHD dataset. ClinicalBERT, being well-suited for medical tasks, and DistilBERT, known for its versatility across NLP tasks, served as strong baselines. LastBERT demonstrated competitive performance, validating its design as an efficient and adaptable model for a variety of applications.

This research not only pushes the boundaries of current knowledge in NLP-based mental health diagnostics but also provides a foundation for future studies to build upon and improve the detection and classification of various mental health disorders using automated, scalable methods. The remainder of this paper is structured as follows, Section 2 discusses

the relevant literature on knowledge distillation techniques and ADHD-related text classification, providing the background and context for this study. Section 3 presents the methodology, detailing the knowledge distillation process, model configurations, datasets preparation, and the evaluation metrics used for benchmarking LastBERT's performance. In Section 4, we report the experimental results, including a detailed comparison with Distil-BERT and ClinicalBERT, and demonstrate the robustness of LastBERT across various NLP tasks and its effectiveness in classifying ADHD severity levels. Section 5 offers a discussion of the findings, highlighting the significance of the results, the challenges encountered, shortcomings, and the practical implications of using LastBERT in real-world scenarios. Moreover, Section 6 concludes the paper by summarizing the contributions and outlining potential avenues for future research. Finally, 7 provides the links to the repositories that contain the model, dataset, and code.

## 2 Related works

Knowledge distillation has been a prominent method for model compression, aiming to reduce the size and computational requirements of large neural networks while maintaining their performance. The concept of distillation, as defined by the Cambridge Dictionary, involves extracting essential information and has been adapted into neural networks to transmit knowledge from a large teacher model to a smaller student model [1]. The idea of moving knowledge from a big model to a smaller one using soft targets was first presented in the fundamental work on knowledge distillation by Hinton et al. [2]. This method has been extensively embraced and expanded in many spheres. While newer LLMs like GPT [3] and LLaMA [4] offer impressive generative capabilities, BERT and its distilled variants (e.g., DistilBERT) remain highly efficient and versatile for sentence-level tasks, making them ideal for developing lightweight models through knowledge distillation. BERT (Bidirectional Encoder Representations from Transformers) first presented by Devlin et al. [5] has grown to be a pillar in natural language processing (NLP). Later studies have concentrated on simplifying BERT models to produce more effective variations. Sanh et al. [6] created DistilBERT, a smaller and faster variation of BERT that largely maintains its original performance. Another distilled variant that achieves competitive performance with a smaller model size is TinyBERT [7], by Jiao et al. Emphasizing task-specific knowledge distillation, Tang et al. [8] showed that BERT may be efficiently distilled into simpler neural networks for particular tasks. Further developments comprise multi-task knowledge distillation as used in web-scale question-answering systems by Yang et al. [9] and Patient Knowledge Distillation [10], which gradually transmits knowledge from teacher to student. These techniques underline the adaptability and efficiency of knowledge distillation in many uses. Moreover, other techniques have been explored to enhance model efficiency. For example, Lan et al. [11] introduced ALBERT (A Lite BERT), which reduces model parameters by sharing them across layers. Press et al. [12] and Bian et al. [13] examined the internal mechanisms of transformer models to further optimize their performance and efficiency. In addition to model compression, there has been significant work in analyzing and improving transformer models. Clark et al. [14] and Kovaleva et al. [15] provided insights into the attention mechanisms of BERT, revealing potential areas for optimization. Kingma and Ba [16] introduced the Adam optimizer, which has been widely used in training transformer models. Kim et al. [17] proposed a novel KD framework called Tutor-KD, which integrates a tutor network to generate samples that are easy for the teacher but difficult for the student model. Their approach improves the performance of student models on GLUE tasks by dynamically controlling the difficulty of training examples during pre-training. This framework demonstrated the importance of using adaptive sample generation to enhance

the student model's learning process. Moreover, in recent studies on the subject, Lin, Nogueira, and Yates [18] introduced a multi-level KD approach for BERT models, known as MLKD-BERT. Their work focused on compressing BERT-based transformers for text ranking applications. The research highlights the challenges of using large transformers and demonstrates how multi-level distillation helps retain the performance of compressed models, offering insights into the potential of KD for various NLP tasks. Another work by Kim et al. [19] focused on creating Bat4RCT, a suite of benchmark data and baseline methods for the classification of randomized controlled trials (RCTs). Although not directly related to ADHD classification, this research demonstrates the effectiveness of KD models in text classification, providing evidence of the potential applicability of KD techniques across different domains. In their subsequent research, Kim et al. [20] explored LERCause, a framework leveraging deep learning for causal sentence identification in nuclear safety reports. The study presents a compelling application of NLP techniques to real-world datasets, reinforcing the value of lightweight, distilled models in complex text analysis tasks. The findings from this study align with our work by showing how KD models can be deployed effectively for specialized tasks, such as mental health diagnostics.

In the context of ADHD classification, NLP has been employed to detect ADHD symptoms from text data. Malvika et al. [21] applied the BioClinical-BERT model to electronic health records (EHR) datasets. Using NLP techniques, they achieved an F1 score of 0.78, demonstrating the potential of pre-trained clinical language models in mental health diagnosis. Peng et al. [22] developed a multi-modal neural network platform using a 3D CNN architecture. Their model integrated functional and structural MRI data from the ADHD-200 dataset, achieving an accuracy of 72.89%. This multi-modal approach highlights the effectiveness of combining neuroimaging modalities for ADHD diagnosis. Chen et al. [23] adopted a decision tree model for ADHD diagnosis on a clinical dataset. Their machine learning approach achieved an accuracy of 75.03%, leveraging interpretable rules to assist healthcare professionals in decision-making processes. Alsharif et al. [24] combined machine learning and deep learning techniques, utilizing Random Forest models on Reddit ADHD data. They achieved an accuracy of 81%, demonstrating the applicability of social media data for identifying ADHD symptoms. Cafiero et al. [25] employed a Support Vector Classifier (SVC) on self-defining memory datasets. Their model achieved an F1 score of 0.77, highlighting the use of memory-based features for mental health classification. Lee et al. [26] applied RoBERTa, a transformer-based language model, on data from ADHD-related subreddits. Their NLP approach achieved an accuracy of 76%, showing the effectiveness of transformer models in identifying ADHD symptoms from social media content. Recent studies on ADHD and NLP have made significant strides in understanding and classifying ADHD-related concerns. A preprint study from MedRxiv examined neurobiological similarities between ADHD and autism, shedding light on age- and sex-specific cortical variations that influence diagnosis and overlap between these conditions. This research emphasizes the importance of text-based approaches in identifying behavioral patterns associated with ADHD [27]. In addition, Lin et al. (2023) explored the role of pretrained transformer models, such as BERT, in healthcare and biomedical applications, with specific emphasis on text-ranking tasks. Their study highlights the potential of NLP-based tools in mental health assessments, including ADHD diagnostics from textual data [28].

These recent studies have shown that advanced NLP techniques and machine learning models are valuable tools for ADHD detection. This body of research highlights the potential of using distilled models and NLP techniques to develop efficient and effective systems for mental health assessment.

# 3 Methodology

This research emphasizes the creation of a compact and efficient student BERT model through knowledge distillation from a pre-trained BERT large model, followed by the evaluation of GLUE benchmark datasets, and the potency of the produced student model, LastBERT [29], on a mental health dataset. This was assessed to demonstrate its practical application by performing ADHD severity classification. The methodology is divided into three main parts:

3.1 The student model formation using the knowledge distillation method.

3.2 Testing robustness of LastBERT model on six GLUE benchmark datasets.

3.3 Performing ADHD severity classification using the LastBERT model.

## 3.1 Student model formation using knowledge distillation

**3.1.1 Teacher and student model configuration.**   The knowledge distillation The procedure starts with the selection of the teacher model. In this research, we employed the pre-trained BERT large model (*bert-large-uncased*) as the teacher model for its remarkable performance across various NLP tasks. Comprising 24 layers, 16 attention heads, and a hidden size of 1024 that is, almost 340 million parameters. This significant capacity makes the BERT large model a great source for knowledge distillation since it helps it to capture complex patterns and relationships in the data [30].

On the other hand, the student model is a customized, smaller variation of the BERT base model (*bert-base-uncased*) designed to maintain efficiency by reducing complexity. This model is configured with 6 hidden layers, 6 attention heads, a hidden size of 384, and an intermediate size of 3072. The setup aligns with insights but not exactly from Jiao et al. [7], which emphasized the importance of optimizing hidden layers and attention heads to balance efficiency and performance in smaller models. Additionally, Sanh et al. [6] demonstrated that compact student models can achieve near-teacher performance by fine-tuning these architectural elements. To distinguish it from other BERT variants, the student model is named LastBERT. These optimizations ensure that LastBERT maintains competitive performance while significantly reducing model size and computational demands. Compared to the 110 million parameters of the BERT base model [31], the student model comprises only 29.83 million parameters. This lightweight architecture, defined with *BertConfig* and instantiated as *BertForMaskedLM*, allows for efficient deployment in resource-constrained environments. Table 1 provides a detailed overview of the hyperparameter settings used in the LastBERT model.

The BERT large model's outstanding performance across a broad spectrum of NLP tasks is attributed to its higher capacity and deeper design, which helps it to learn more complicated representations of the data, which is the main reason why it is chosen as the teacher model. By extracting knowledge from the BERT large model, we aim to transfer this high level of understanding to a more compact and efficient student model. While still gaining from the distilled expertise of the BERT large model, the specific configuration of the student model guarantees that it is considerably smaller and faster, hence more suitable for deployment in resource-limited surroundings. The distillation process centers on moving the knowledge from the last layer of the instructor model to the student model. This method uses the thorough knowledge included in the last representations of the instructor model, which captures the condensed knowledge from all past layers.

Transformers, which serve as the foundation for models like BERT, employ multi-head attention mechanisms. This enables the model to simultaneously focus on different parts of

**Table 1. Configuration of the LastBERT model.**

| Parameter | Value |
| --- | --- |
| architectures | BertForMaskedLM |
| attention_probs_dropout_prob | 0.1 |
| classifier_dropout | null |
| gradient_checkpointing | false |
| hidden_act | gelu |
| hidden_dropout_prob | 0.1 |
| hidden_size | 384 |
| initializer_range | 0.02 |
| intermediate_size | 3072 |
| layer_norm_eps | 1e-12 |
| max_position_embeddings | 512 |
| model_type | bert |
| num_attention_heads | 6 |
| num_hidden_layers | 6 |
| pad_token_id | 0 |
| position_embedding_type | absolute |
| transformers_version | 4.42.4 |
| type_vocab_size | 2 |
| use_cache | true |
| vocab_size | 30522 |
| **Number of Parameters: 29 Million** | |

the input sequence. The multi-head attention mechanism can be described as:

$$\text{MultiHead}(Q, K, V) = \text{Combine}(\text{head}_1, \text{head}_2, \ldots, \text{head}_n) W^O \tag{1}$$

where each individual attention head is computed as:

$$\text{head}_m = \text{Attention}(Q W_m^Q, K W_m^K, V W_m^V) \tag{2}$$

In this formulation, the query, key, and value matrices are $Q$, $K$, and $V$ respectively. Specific to each head $m$ the projection matrices $W_m^Q$ $W_m^K$ and $W_m^V$ are $W^O$ and $W_m^V$ respectively. Every head generates concatenated, linearly transformed outputs from a distinct learned linear projection of these matrices.

Usually, one defines the attention mechanism itself as:

$$\text{Attention}(Q, K, V) = \text{softmax}\left(\frac{Q K^T}{\sqrt{d_k}}\right) V \tag{3}$$

Where $d_k$ is the dimensionality of the key vectors functioning as a scaling mechanism. One guarantees with the softmax function the sum of the attention weights.

This method allows every attention head to focus on different sections of the input sequence, therefore capturing many aspects of the interactions between words or tokens in the sequence [32]. The architectural diagram represented in Fig 1 gives us detailed insights and layers of the student model.

**3.1.2 Rationale for model selection.** The choice of BERT-based models stems from their proven efficiency across various NLP tasks, such as text classification, sentiment analysis, and question answering. BERT offers an optimal balance between performance and computational

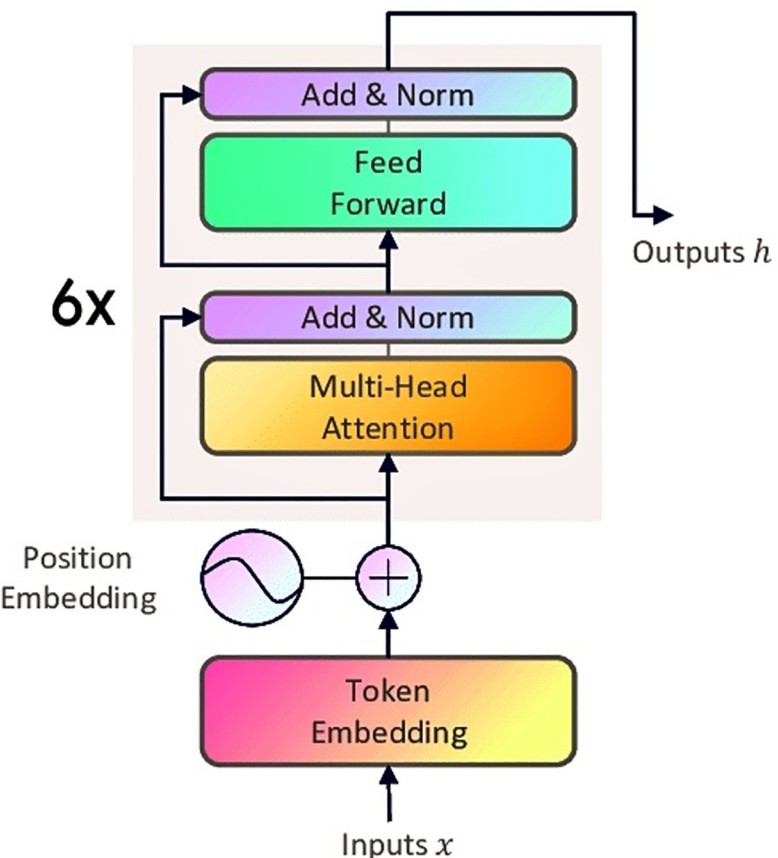

**Fig 1. Student model architecture.**

feasibility, making it suitable for knowledge distillation. While advanced LLMs like GPT [3] and LLaMA [4] demonstrate state-of-the-art capabilities, their large size-—often billions of parameters, make them impractical for training or fine-tuning on resource-constrained platforms like Colab and Kaggle, both limited to 16GB VRAM and short session times. In contrast, BERT and its distilled versions, including DistilBERT [6] and TinyBERT [7], are more practical for researchers. These models offer competitive performance while being accessible for fine-tuning with limited resources. The objective of this research is to develop LastBERT-a compact, efficient model that performs well across NLP tasks and can be easily fine-tuned on free platforms. This empowers developers and researchers to leverage LastBERT for various NLP tasks without the need for extensive computational resources.

**3.1.3 Dataset preparation.** The distillation process benefited much from the WikiText-2-raw-v1 dataset. Suitable for language modeling initiatives, this collection offers a spectrum of text samples [33]. The dataset was tokenized using the Hugging Face library's *Bert Tokenizer*. Tokenizing guaranteed constant input dimensions for both the teacher and student models by padding and restricting inputs to a maximum length of 128 tokens. This preprocessing stage keeps consistency in model inputs and helps to enable efficient information flow during the distillation process.

**3.1.4 T3raining procedure.** Using the Kullback-Leibler divergence loss function (*KLDiv-Loss*), the knowledge distillation process fundamentally conveys the knowledge from the teacher BERT Large model to the student LastBERT model. This loss function gauges how

different the probability distributions produced by the student BERT model and the teacher BERT model. Between two probability distributions $X$ and $Y$ the Kullback-Leibler (KL) divergence is defined as:

$$D_{\mathrm{KL}}(X \parallel Y) = \sum_i X(i) \log\left(\frac{X(i)}{Y(i)}\right)$$ (4)

Within the framework of knowledge distillation, the softened probabilities are computed as:

$$p_i^T = \mathrm{softmax}\left(\frac{z_i^T}{T}\right)$$ (5)

$$p_i^S = \mathrm{softmax}\left(\frac{z_i^S}{T}\right)$$ (6)

The KL divergence loss function for knowledge distillation is:

$$\mathcal{L}_{\mathrm{distill}} = -\frac{1}{N}\sum_{i=1}^{N}\sum_{j=1}^{C} p_{ij}^T \log\left(\frac{p_{ij}^S}{p_{ij}^T}\right)$$ (7)

The combined loss function used in the training is:

$$\mathcal{L} = \alpha \cdot T^2 \cdot \mathcal{L}_{\mathrm{distill}} + (1 - \alpha) \cdot \mathcal{L}_{\mathrm{CE}}$$ (8)

Initially, the AdamW optimizer was applied with a learning rate of $5 \times 10^{-5}$ [34]. The AdamW optimization algorithm is an extension of the Adam algorithm to correct the weight decay. The update rule for the weights $w$ in the AdamW optimizer is given by:

$$w_{t+1} = w_t - \eta\left(\frac{\hat{m}_t}{\sqrt{\hat{v}_t} + \epsilon} + \lambda w_t\right)$$ (9)

To enhance training efficiency and model convergence, a learning rate scheduler (*get_linear_schedule_with_warmup*) was implemented. This scheduler modifies the learning rate during the training process, beginning with a warm-up phase and then transitioning to a linear decay. Mixed precision training was utilized to improve computational efficiency [35]. This approach leverages the capabilities of modern T4 GPU to perform computations in both 16-bit and 32-bit floating point precision, reducing memory usage and speeding up training without compromising model accuracy.

The training process was conducted over 10 epochs, with each epoch consisting of 4590 iterations, resulting in a total of 45,900 iterations. Throughout each epoch, the model's performance was assessed on both the training and validation sets. Important performance metrics, including loss, accuracy, precision, recall, and F1 score, were recorded to monitor the model's progress and ensure effective knowledge transfer. The training was performed on the Google Colab notebook service, utilizing T4 GPU to accelerate the computational process [36].

**3.1.5 Algorithm: Knowledge distillation for LastBERT model.** Algorithm 1 describes the knowledge distillation process for training the LastBERT model (student) using the BERT large (teacher) model's predictions. By minimizing both the distillation loss and the cross-entropy loss, the goal is to transfer knowledge from the teacher model to the student model. This approach aids in training a smaller, more efficient model (student) that approximates the performance of the larger model (teacher).

The process involves computing logits from both the teacher BERT large model and student LastBERT model, calculating the softened probabilities using a temperature parameter $T$ with value 2.0, and then computing the distillation and cross-entropy losses. These losses are combined using a weighted sum defined by $\alpha$ to form the total loss. Afterward, it is then back-propagated to update the student model's parameters. The training is carried out in 10 epochs, iterating over mini-batches of the training data until the student model is adequately trained.

**Algorithm 1** Knowledge Distillation for Student Model (LastBERT)

```
 1: Input: Teacher model T, Student model S
 2: Input: Training data D = {(xᵢ, yᵢ)}ᵢ₌₁ᴺ
 3: Input: Temperature T, Distillation loss weight α, Cross-entropy
    loss weight (1 - α), Learning rate η
 4: Output: Trained Student model S
 5: procedure KNOWLEDGEDISTILLATION(T, S, D, T, α, η)
 6:   for each epoch do
 7:     for each minibatch B ⊂ D do
 8:       Forward pass: Compute teacher's logits zᵢᵀ = T(xᵢ) for xᵢ ∈ B
 9:       Forward pass: Compute student's logits zᵢˢ = S(xᵢ) for xᵢ ∈ B
10:       Compute softened probabilities pᵢᵀ = softmax(zᵢᵀ/T)
11:       Compute softened probabilities pᵢˢ = softmax(zᵢˢ/T)
12:       Compute distillation loss using KL divergence:
```

$$\mathcal{L}_{\mathrm{distill}} = \frac{T^2}{|B|} \sum_{i \in B} \mathrm{KL}(p_i^T \parallel p_i^S)$$

```
13:       Compute cross-entropy loss:
```

$$\mathcal{L}_{\mathrm{CE}} = -\frac{1}{|B|} \sum_{i \in B} y_i \log(\mathrm{softmax}(z_i^S))$$

```
14:       Combine losses:
```

$$\mathcal{L} = \alpha \cdot \mathcal{L}_{\mathrm{distill}} + (1 - \alpha) \cdot \mathcal{L}_{\mathrm{CE}}$$

```
15:       Backward pass: Compute gradients of L w.r.t. S's parameters
16:       Update S's parameters using optimizer (e.g., Adam) with
          learning rate η
17:     end for
18:   end for
19:   return trained Student model S
20: end procedure
```

**3.1.6 Model saving.**   For ensuring persistence and enable further evaluation, the student model was saved after the distillation process. The model-saving process was automated and integrated with Google Drive for efficient storage management. This approach ensures that intermediate models are preserved, allowing for retrospective analysis and comparison. Furthermore, the model has been uploaded to the Hugging Face models repository [29] for easy implementation by developers and researchers using the API.

**3.1.7 Top-level overview of the proposed knowledge distillation approach between student and teacher BERT models.**   Knowledge distillation process between a teacher model (BERT Large Uncased) and a student model (BERT Base Uncased/LastBERT) using the Wiki-Text-2-raw transfer set. The teacher model, pre-trained on masked language modeling (MLM), generates *soft labels* by applying a softmax function at a high temperature *T*. The student model learns by mimicking these soft predictions and also computing *hard predictions* against the ground truth labels. The training process minimizes a weighted sum of the *distillation loss* (between teacher and student soft predictions) and the *student loss* (between student predictions and ground truth). The hyperparameter $\alpha$ balances the contributions of these two

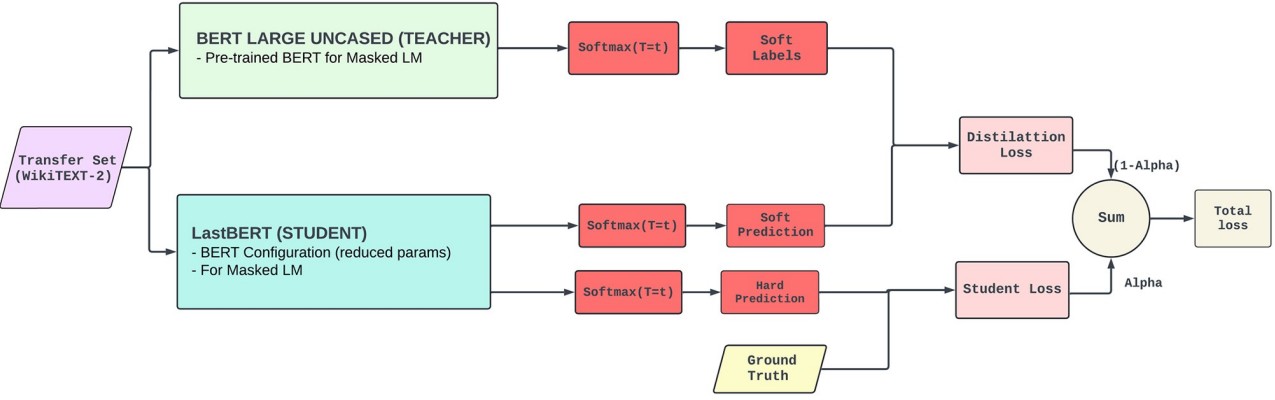

**Fig 2. Top-level overview of the teacher-student knowledge distillation process.**

losses. This setup enables the student model to retain performance comparable to the teacher while significantly reducing the model size and computational cost. The detailed top-level overview of our approach is presented in Fig 2.

## 3.2 Testing on GLUE benchmark datasets

Using General Language Understanding Evaluation (GLUE) benchmark datasets [37], the efficacy of the distilled student model (LastBERT) was evaluated. These datasets include a complete collection of NLP challenges meant to test many facets of language comprehension. For every one of the six GLUE benchmark datasets, the review process consisted of meticulous procedures.

**3.2.1 Microsoft Research Paraphrase Corpus (MRPC).** Aimed at ascertaining if two statements are paraphrases of each other, this binary classification challenge [38]. Using the Hugging Face library [39], the dataset was loaded and tokenized using the *BertTokenizerFast*, therefore guaranteeing consistent input dimensions via padding and truncation. After that, our LastBERT model with six attention heads, six hidden layers, a hidden size of 384, and an intermediary size of 3072 was tuned on this dataset. The training consisted of two epochs using an 8-batch size and a learning rate of $5 \times 10^{-5}$. The evaluation schedule called for looking at every epoch. We computed performance measures including accuracy, precision, recall, and F1 score to evaluate the model's performance in paraphrase identification.

**3.2.2 Stanford Sentiment Treebank (SST-2).** Another binary classification challenge is to classify the positive or negative essence of any sentence by doing sentiment analysis using this dataset [40]. The dataset was loaded and tokenized, much like MRPC's. To improve the training set, synonym replacement was also used for data augmentation. With early halting enabled, batch size of 8, and a weight decay of 0.01 applied for regularity, the same LastBERT model configuration was utilized and refined across 10 epochs with a learning rate of $2 \times 10^{-5}$. Accuracy, precision, recall, and F1 score let one assess the model's performance. To show the training development, these measures were entered into logs and plotted.

**3.2.3 Corpus of Linguistic Acceptability (CoLA).** Inspired by sentiment analysis, this dataset [41] reveals still another binary classification difficulty. The dataset was loaded and tokenized just as with MRPC's. Synonym replacement also was utilized for data augmentation to enhance the training set. Early stopping enabled, batch size of 8, and a weight decay of 0.01 applied for regularity, the same LastBERT model configuration was used and refined across 10

epochs with a learning rate of $2 \times 10^{-5}$. One can evaluate the model by means of accuracy, precision, recall, and F1 score. These events entered logs and were plotted to indicate the evolution of training.

**3.2.4 Quora Question Pairs (QQP).**  The QQP dataset consists of determining whether two questions have semantically comparable meanings [42]. By tokenizing the dataset, one may guarantee constant input lengths by truncation and padding. The LastBERT model was refined during three epochs using a batch size of sixteen and a learning rate of $2 \times 10^{-5}$. We applied $1 \times 10^{-2}$ weight decay during 500 warm-up steps. The evaluation criteria consisted of accuracy, precision, recall, and F1 score, thereby offering a whole picture of the model's performance in identifying question equivalency.

**3.2.5 Multi-Genre Natural Language Inference (MNLI).**  Establishing the link between a premise and a hypothesis is necessary for this multi-class classification task [43]. Tokenizing the information ensured consistency. Refined throughout three epochs, the LastBERT model had an 8-batch size and a learning rate of $5 \times 10^{-5}$. The evaluation focused on accuracy, thereby providing a clear performance standard for the model in natural language inference challenges.

**3.2.6 Semantic Textual Similarity Benchmark (STS-B).**  This dataset comprises semantic similarity between two phrases assessing task using regression [44]. Tokenizing the dataset allowed constant input dimensions over the model to be maintained. Over six epochs, the LastBERT was maximized with an 8,000 batch size and a learning rate of $5 \times 10^{-5}$. Performance was assessed using Pearson and Spearman correlation coefficients to ascertain how exactly the model could predict semantic similarity.

**3.2.7 Computational setup.**  All training and evaluation were conducted using cloud-based platforms, specifically Google Colab with T4 GPU and Kaggle Notebooks with P100 GPU. These platforms provide cost-effective access to high-performance computing resources, enabling efficient model training and experimentation without requiring dedicated hardware. Such an environment ensures that the experiments are reproducible and accessible to researchers working with limited computational resources. The *Trainer* class from the Hugging Face library was employed to manage the training and evaluation pipelines effectively. Custom callbacks were implemented for logging key metrics and saving the best-performing models during training. After each training run, metrics such as accuracy, precision, recall, and F1 score were extracted and visualized to gain insights into the model's performance. The evaluation results were compared with other baseline models, such as BERT base, BERT large, TinyBERT, MobileBERT, and DistilBERT to highlight trade-offs between model performance and size, demonstrating the practicality and efficiency of the distilled student model (LastBERT) for real-world applications.

## 3.3 Application on mental health dataset

To extend the applicability of LastBERT beyond standard NLP tasks, we evaluated its performance on the Reddit Mental Health Dataset (RedditMH) [45], sourced from Hugging Face [46]. This dataset consists of over 151,000 posts covering various mental health conditions, such as ADHD, Depression, OCD, and PTSD. We focused specifically on posts related to ADHD to assess LastBERT's ability to classify user concerns by severity. Given the importance of identifying nuanced mental health symptoms, this dataset serves as a valuable benchmark for evaluating NLP models in real-world diagnostic settings.

**3.3.1 Data preparation.**  To prepare the dataset, we filtered posts exclusively from the ADHD subreddit. The "title" and "body" fields were combined into a single "text" field to create a unified input for the model. Any missing values were replaced with empty strings, and irrelevant metadata—such as "author," "created_utc," and "url"—was removed. Posts were

**Table 2. Sample Posts from the ADHD dataset.**

| Score | Severity | Title | Post |
|---|---|---|---|
| 1 | Mild | is there a way i can make it so my computer only does schoolwork | is there a way i can make it so my computer only does schoolwork im trying to do schoolwork right now but i keep getting distracted with reddit, i just got my adderall prescription, but for some reason im like hyperfocused on reddit right now and i want to make it so i cant go on any "fun" sites does anyone know how i could do this? i have a thinkpad windows laptop btw |
| 5 | Severe | Why is getting diagnosed as an adult so freaking hard? | My wife's therapist recently suggested she read a book on women with ADHD to see if it tracked with her. To be supportive I started reading along. Now that I'm diving into ADHD I'm seeing my entire life, all my struggles and failures laid out in front of me. Well, my wife's therapist left the practice and my wife can't find anyone to screen her for ADHD. She's called dozens of places, followed so many leads. All for nothing. No one screens for adults. I'm talking to my doctor and he put in a referral but the place only tests kids. They said we'd probably have to go (hours away) to Stanford (if they'd even do it for us) and pay out of pocket, which for the two of us would be between $6k and $10k. We have enough trouble following through on anything without having to go on multiple wild goose chases just to get help. It makes me want to scream. Talking to my mom, and my son sounds just like me at 4. At least he can probably get tested, right? When do I even do that? |
| 2 | Mild | Lightheaded, almost like a bobble head feeling when not on medication (concerta)? | Lightheaded, almost like a bobble head feeling when not on medication (concerta)? I've been taking concerta for a couple months now but I've started noticing, days when I'm not on it, I have a literal lightheaded feeling, like my head is really light and I can't really keep it straight. Idk it's hard to explain. I don't feel dizzy and it doesn't effect me really it's just annoying and feels weird |

assigned severity labels: "Mild" for scores less than or equal to 2 and "Severe" for scores greater than 2. The outliers were handled by capping scores between 0 and 5 to ensure consistency. Moreover, the class imbalance was addressed by upsampling the minority class (Mild) to match the majority class (Severe), resulting in two balanced classes of 14,726 entries each. The dataset was then divided into training (80%) and test (20%) sets to ensure balanced class representation throughout the process. The final dataset consisted of 29,452 records, ready for training and evaluation. This preprocessing ensured clean, structured data suitable for assessing the model's performance in classifying ADHD-related concerns. The Table 2 demonstrates three examples of ADHD-related concerns posts on Reddit with severity levels.

**3.3.2 Tokenization.** The Hugging Face library's *Bert TokenizerFast* tokenized the textual data. This tokenizer was chosen for its speed and efficiency in managing vast amounts of data as well as for being the same tokenizer used to build the student model, LastBERT. Padding and truncating the inputs to a maximum length of 512 tokens guaranteed that all inputs matched a consistent size fit for BERT-based models. Maintaining the performance and compatibility of the model depends on this last stage. Labels were also included in the tokenized dataset to support supervised learning, therefore guaranteeing that the model could learn from the given samples efficiently.

**3.3.3 Model configuration and training.** This work used the LastBERT model, generated from the knowledge distillation process utilizing the pre-trained BERT big model. With a hidden size of 384, six attention heads, six hidden layers, and an intermediary size of 3072 the model's configuration was especially designed to strike efficiency and performance. While preserving a good degree of performance, this arrangement greatly lowers computational needs. The training was carried out with the Hugging Face library's *Training Arguments* class. Important values were a learning rate of $2 \times 10^{-5}$, a batch size of 8 for both training and evaluation and a 10-epoch training duration. Furthermore added to prevent overfitting and enhance generalization was a weight decay of $1 \times 10^{-2}$. Early halting was used to halt instruction should the performance of the model not show improvement over three consecutive assessment periods. Operating in a GPU environment, the training method took advantage of the student model's efficient design to properly manage the computing demand.

**3.3.4 Evaluation metrics.** To present a whole picture of the model's performance, several measurements were taken—accuracy, precision, recall, and F1 score—all computed using the *evaluate* library from Hugging Face. These metrics were kept track of all through the training process to regularly monitor the evolution of the model. To enable thorough research and guarantee that any performance problems could be quickly resolved, a special callback class, *MetricsCallback*, was used to methodically record these measurements. The model was assessed on the test set to produce predictions upon training completion. Then, by matching these projections with the real labels, one may assess the model's performance. Moreover, a confusion matrix was developed to demonstrate the performance of the model over various classes, therefore providing knowledge about its classification ability. The confusion matrix was exhibited using Seaborn's heatmap tool stressing the number of true positive, true negative, false positive, and false negative forecasts. By exposing in-depth information on the model's ability to properly identify postings linked with ADHD, this testing method underscored the likely advantages of the model in mental health diagnosis. Particularly in the field of mental health, the continuous performance of the model over several criteria exposed its dependability and adaptability for use under pragmatic environments.

**3.3.5 Algorithm: ADHD classification using LastBERT model.** The Algorithm 2 describes this paper's training and evaluation of the LastBERT model on the ADHD dataset from Reddit Mental Health forums. The procedure begins with dataset preparation by filtering ADHD posts, concatenating "title" and "body," creating labels, and capping scores. Then, upsampling the minority class helps to balance the dataset and split it into test and training groups. A BERT tokenizer tokenizes the text data and then truncates it to 512 tokens. Initializing the training parameters—batch size, learning rate, epochs, early termination tolerance—is simple. During training, the LastBERT model's logits are computed for each minibatch, and the cross-entropy loss is calculated and used for the backward pass and parameter updates. The model is evaluated on a validation set after each epoch, with early stopping applied if there is no improvement. After training, the final evaluation is conducted on the test set. Metrics such as accuracy, precision, recall, and F1 score are reported. A classification report and confusion matrix are generated to offer a comprehensive evaluation of the model's performance. The trained LastBERT model is then returned.

**Algorithm 2** ADHD Classification Using LastBERT Model

```
 1: Input: Reddit Mental Health dataset D, LastBERT model S
 2: Input: Tokenizer T, Learning rate η, Batch size B, Epochs E, Early
      stopping patience P
 3: Output: Trained LastBERT model S
 4: procedure TRAINANDEVALUATE(D, S, T, η, B, E, P)
 5:   Preprocess dataset: Filter ADHD posts, concatenate 'title' and
        'body' to 'text', cap scores, define labels
 6:   Balance dataset by upsampling the minority class
 7:   Split dataset into training and test sets (80:20)
 8:   Tokenize text data with T, truncate to 512 tokens
 9:   Initialize training parameters: batch size B, learning rate η,
        epochs E, early stopping patience P
10:    for epoch e ∈ E do
11:      for minibatch b ⊂ train_dataset do
12:        Compute LastBERT logits z_i = S(x_i) for x_i ∈ b
13:        Compute cross-entropy loss L_CE
14:        Backward pass and parameter update
15:      end for
16:      Evaluate on validation set
17:      Apply early stopping if no improvement
18:    end for
```

```
19:   Final evaluation on test set
20:   Compute and report metrics: accuracy, precision, recall, F1
      score
21:   Generate classification report and confusion matrix
22:   return trained LastBERT model S
23: end procedure
```

**3.3.6 Top-level overview of the proposed ADHD classification using LastBERT and other models.** Fig 3 portrays the workflow of the ADHD severity classification training approach. It illustrates the two main phases: *Data Augmentation* and *Training*. In the Data

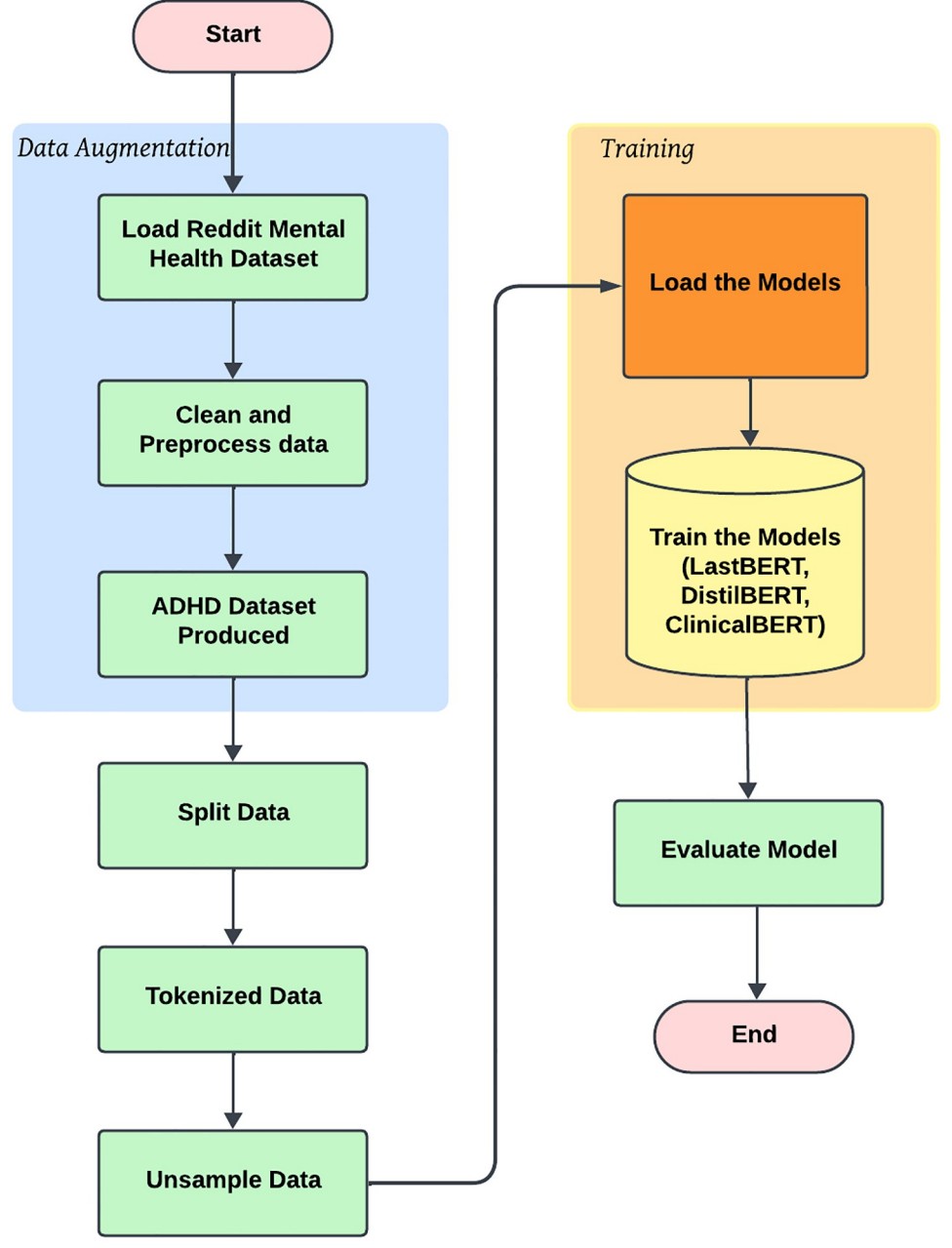

**Fig 3. Top-level overview for ADHD classification study.**

Augmentation phase, the Reddit Mental Health dataset is loaded, followed by cleaning and preprocessing steps to generate the ADHD dataset. The data is then split, tokenized, and balanced using unsampling techniques. The Training phase involves loading and training models like LastBERT, DistilBERT, and ClinicalBERT on the prepared dataset. The final step includes model evaluation to assess performance, marking the end-to-end ADHD classification workflow.

## 4 Results

In this section, we present a comprehensive evaluation of the LastBERT model and its comparison with other knowledge distillation models like BERT large, BERT base, TinyBERT, DistilBERT, and MobileBERT. The results are based on various natural language processing (NLP) tasks using datasets from the GLUE benchmark. We analyze the model's performance metrics, such as accuracy, precision, recall, and F1 score, across different tasks including paraphrase identification, sentiment analysis, text classification, and grammatical acceptability. Additionally, we compare the performance of LastBERT in the context of ADHD-related text classification with ClinicalBERT and DistilBERT, providing insights into its efficacy in resource-constrained environments. Through this detailed analysis, we assess how the knowledge distillation process contributes to creating an efficient, compact model without significant compromises in performance.

### 4.1 Knowledge distillation process

The knowledge distillation process, conducted over 10 epochs, demonstrated a steady improvement in performance metrics, including loss, accuracy, precision, recall, and F1 score shown in Fig 4. The training and validation loss curves indicate a consistent decrease in the loss for both datasets, with an initial sharp decline that stabilizes as the epochs progress. This trend suggests effective learning without significant overfitting, as evidenced by the validation loss not increasing. Accuracy metrics show a rapid improvement in the early epochs, followed by a plateau, indicating that the model has reached a stable performance level. The validation

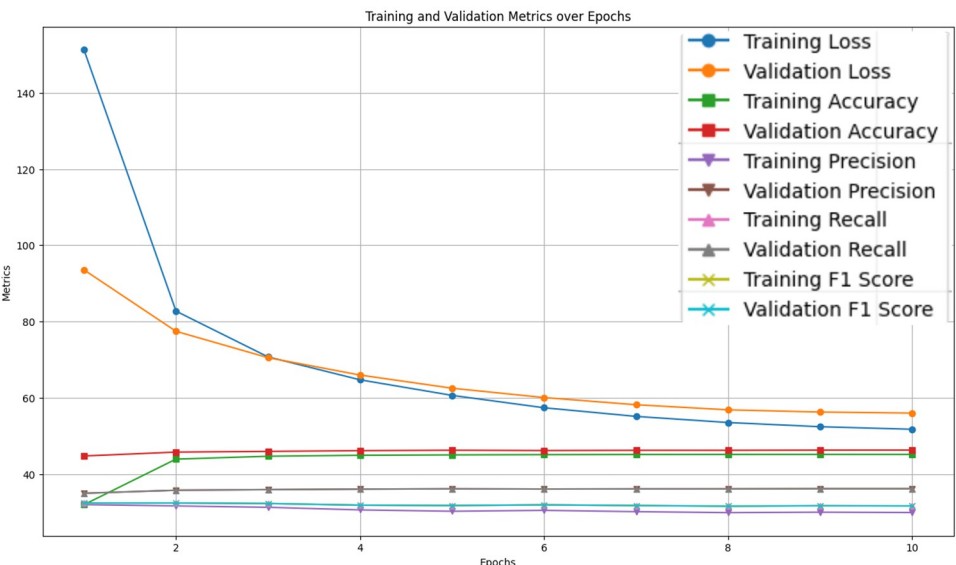

**Fig 4. Training and validation metrics over Epochs during Knowledge distillation process.**

accuracy closely follows the training accuracy, further confirming the absence of overfitting. Precision, recall, and F1 score metrics exhibit similar patterns, with initial rapid increases followed by stabilization, demonstrating that the model's ability to correctly identify relevant instances and its capability to identify all relevant instances have both improved and stabilized over time. The F1 score, representing the harmonic mean of precision and recall, consistently maintains balance throughout the training process.

## 4.2 Evaluation on GLUE benchmark datasets

The distilled student model, LastBERT was evaluated on multiple GLUE benchmark datasets to assess its performance across various NLP tasks. Table 3 summarizes the best performance metrics achieved for each dataset.

The LastBERT model's performance on various GLUE benchmark datasets indicates its robustness across different natural language processing tasks. On the MRPC dataset, the model achieved an accuracy of 71.08%, precision of 73.07%, recall of 91.40%, and an F1 score of 81.21%, demonstrating its capability in identifying paraphrases with a good balance between precision and recall (Fig 5). For the SST-2 dataset, the model exhibited high effectiveness in sentiment analysis tasks, achieving an accuracy of 82.11%, precision of 81.03%, recall of 84.68%, and an F1 score of 82.82% (Fig 6). The CoLA dataset, which focuses on grammatical acceptability, proved more challenging, with the model achieving an accuracy of 69.13% and a Matthews correlation coefficient of 0.171 where the range is considered -1 to 1 for this metric, indicating reasonable performance in grammatical judgment tasks (Fig 7). On the QQP dataset, the model performed strongly in detecting semantic equivalence between questions, achieving an accuracy of 82.26%, precision of 73.15%, recall of 81.87%, and an F1 score of 77.27% (Fig 8). The MNLI dataset, which tests the model's capability in natural language inference across various genres, saw the model achieving an accuracy of 70.45% (Fig 9). Lastly, the STS-B dataset, which measures semantic similarity between sentence pairs, demonstrated the model's effectiveness with Pearson and Spearman correlations of 0.34 and 0.35, respectively (Fig 10). These results highlight the model's versatility and efficiency in handling a domain of NLP tasks, validating the success of the knowledge distillation process.

## 4.3 ADHD-related concerns classification method

In this study, we focused on classifying the severity level of ADHD-related concerns from text data extracted from the Reddit Mental Health dataset. We compared the performance of three models: our custom student BERT (LastBERT) model, DistilBERT, and ClinicalBERT. The

**Table 3. Performance of LastBERT on GLUE benchmark datasets.**

| Dataset | Epoch | Accuracy | Precision | Recall | F1 Score | Other Metric |
|---|---|---|---|---|---|---|
| MRPC | 2 | 71.08% | 0.7307 | 0.9140 | 0.8121 | - |
| SST-2 | 2 | 82.11% | 0.8103 | 0.8468 | 0.8282 | - |
| CoLA | 10 | 69.13% | - | - | - | 0.1710 (Matthews Corr.) |
| QQP | 3 | 82.26% | 0.7315 | 0.8187 | 0.7727 | - |
| MNLI | 3 | 70.45% | 0.7056 | 0.7048 | 0.7056 | - |
| STS-B | 6 | - | - | - | - | 0.34 (Pearson), 0.35 (Spearman) |

Performance of LastBERT on various GLUE benchmark datasets, including accuracy, precision, recall, F1 score, and other relevant metrics. The "-" symbol indicates that the specific metric was not required or relevant for that particular dataset.

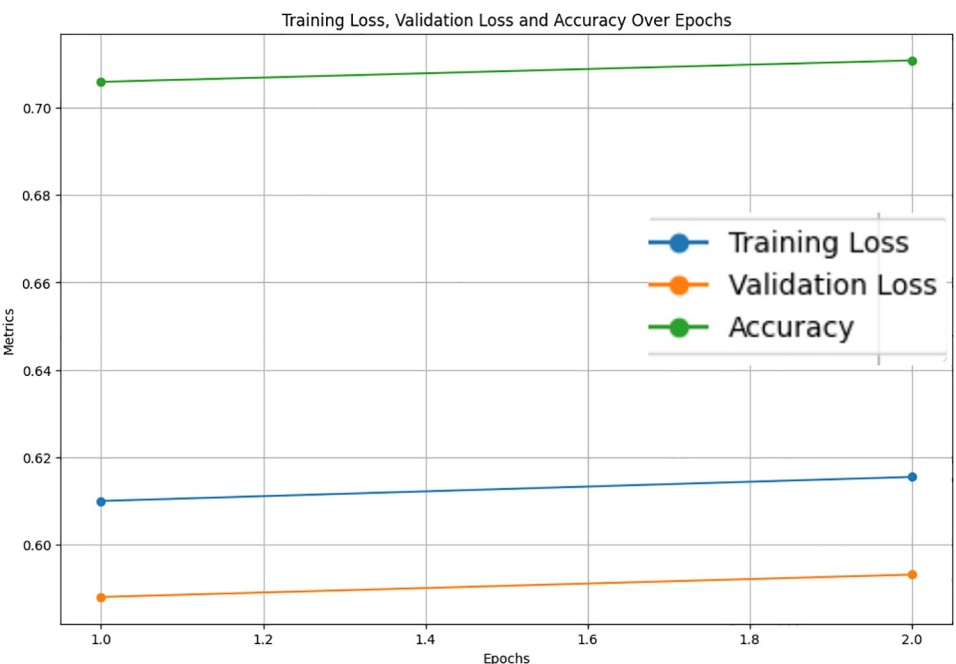

**Fig 5. Accuracy, validation loss, and training loss over Epochs for MRPC.**

models were assessed based on various metrics including precision, recall, F1 score, and accuracy. Below we present a detailed analysis of the results.

**4.3.1 LastBERT model (The distilled student BERT model).** The LastBERT model was evaluated to determine its effectiveness in classifying the severity of ADHD-related concern

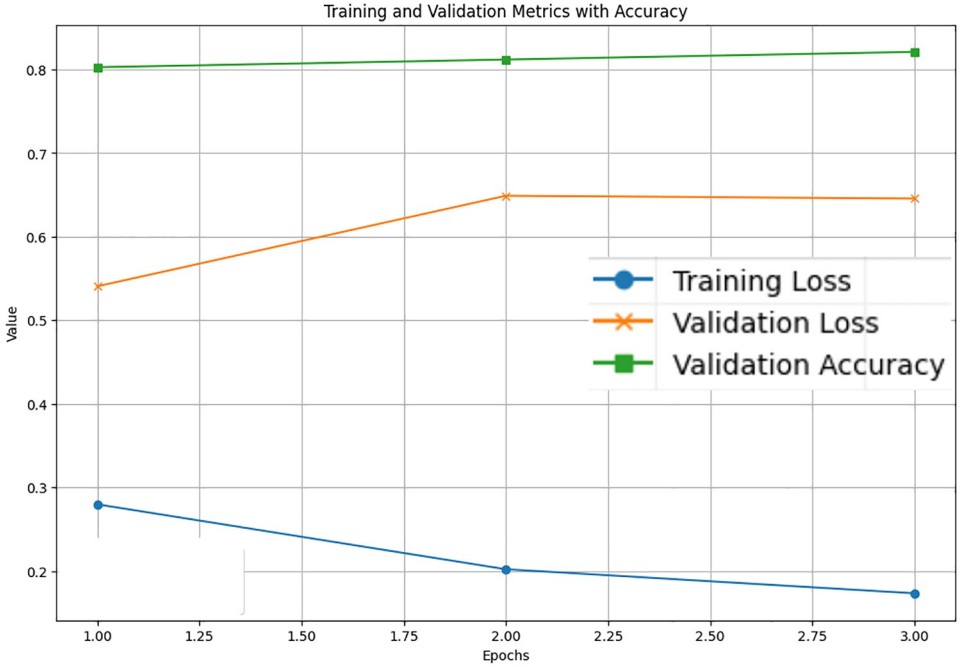

**Fig 6. Accuracy, validation loss, and training loss over Epochs for SST-2.**

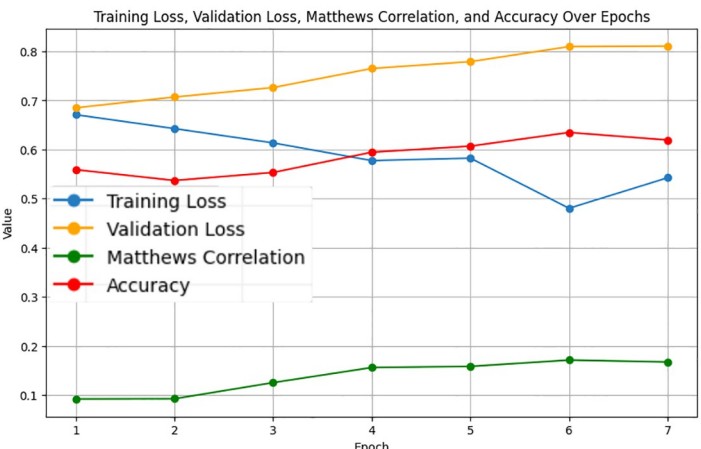

**Fig 7. Validation loss, training loss, Matthews correlation, and accuracy over Epochs for CoLA.**

levels. The model was trained for 1 hour and 33 seconds for 13 epochs. In which the model produced results of 85% accuracy, 85% f1 score, 85% precision, and 85% recall. Fig 11 illustrates the precision, recall, and F1 score over the training epochs. The model shows a steady improvement in these metrics, stabilizing after the initial epochs. Fig 12 displays the accuracy, training loss, and validation loss over epochs, highlighting the model's convergence and stability during training. The training loss consistently decreased, and the validation loss showed minor fluctuations, indicating that the model is well-generalized.

The confusion matrix for the LastBERT model in Fig 13, clearly illustrates the model's performance by showing the number of correctly and incorrectly classified instances. The

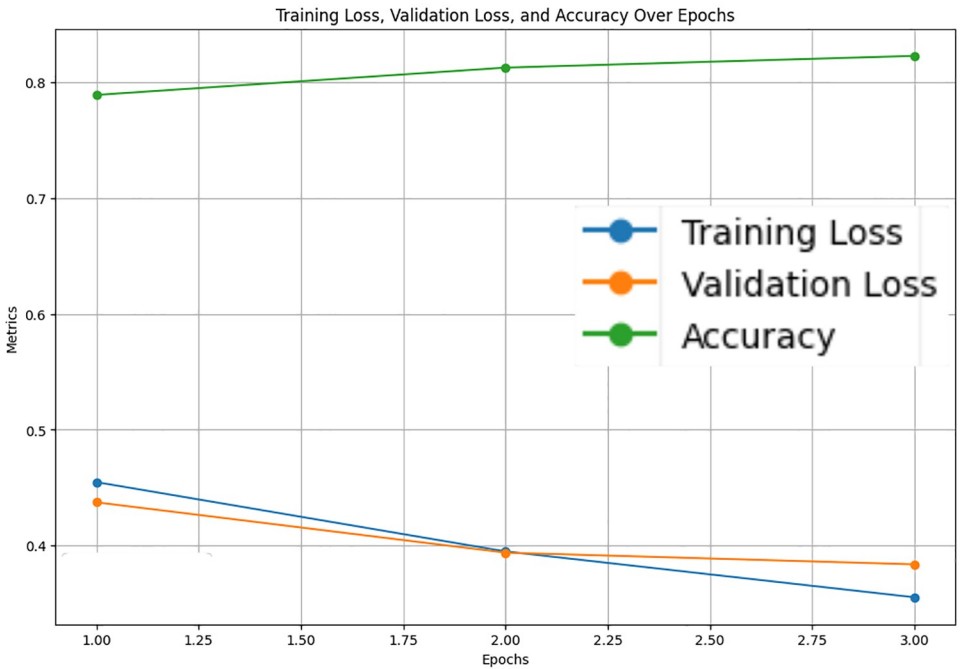

**Fig 8. Accuracy, validation loss and training loss over Epochs for QQP.**

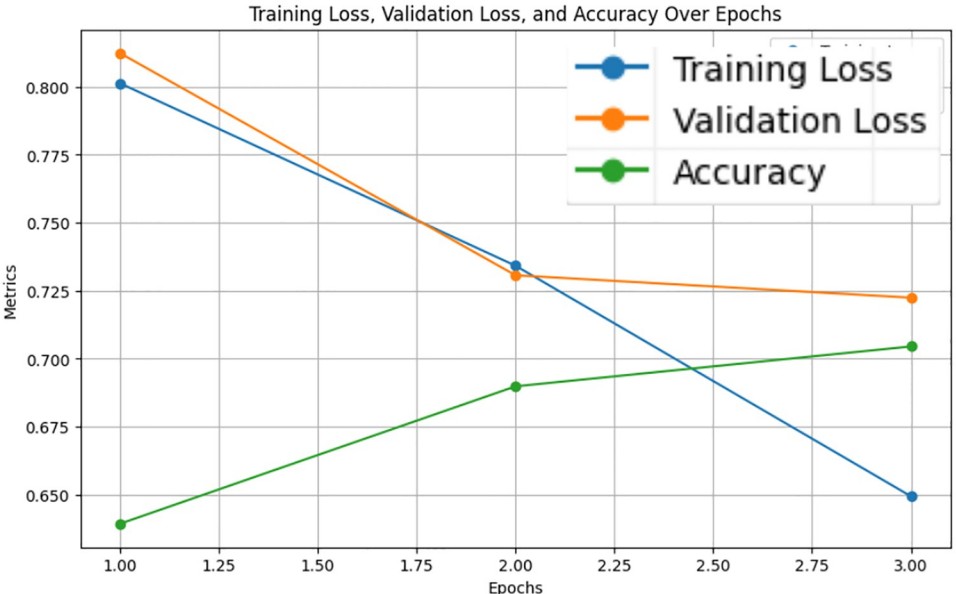

**Fig 9. Accuracy, validation loss, and training loss over Epochs for MNLI.**

LastBERT model achieved an overall accuracy of 85%. Specifically, the confusion matrix indicates that out of 2946 Mild class instances, 2590 were accurately classified, whereas 356 were incorrectly classified as Severe. Similarly, out of 2945 Severe class instances, 2414 were accurately classified, while 531 were mistakenly classified as Mild. These are all unseen test set data

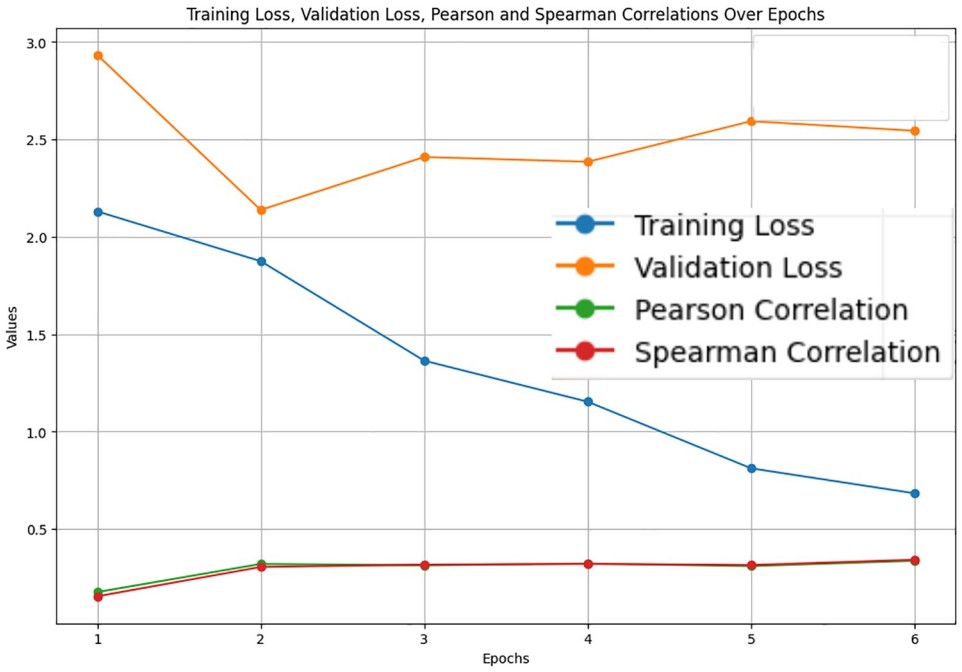

**Fig 10. Validation loss, training loss, Pearson and Spearman correlations over Epochs for STS-B.**

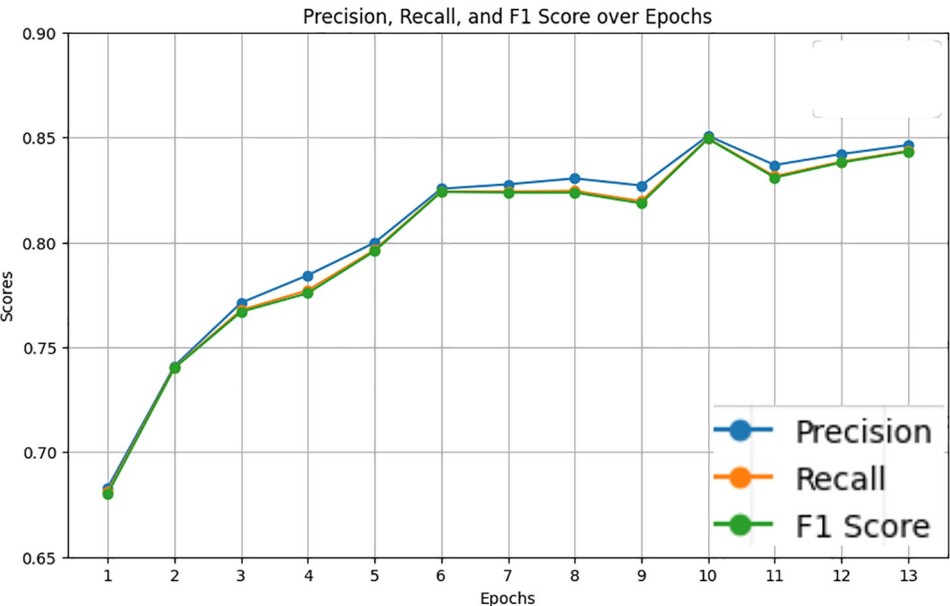

**Fig 11. Precision, recall, and F1 score over Epochs for LastBert model.**

that were not used during the training phase. This balanced performance across both classes highlights the model's ability to effectively distinguish between Mild and Severe ADHD-related severity concerns levels.

The Receiver Operating Characteristic (ROC) curve for the Student BERT model (Fig 14) provides a graphical representation of the true positive rate (sensitivity) against the false

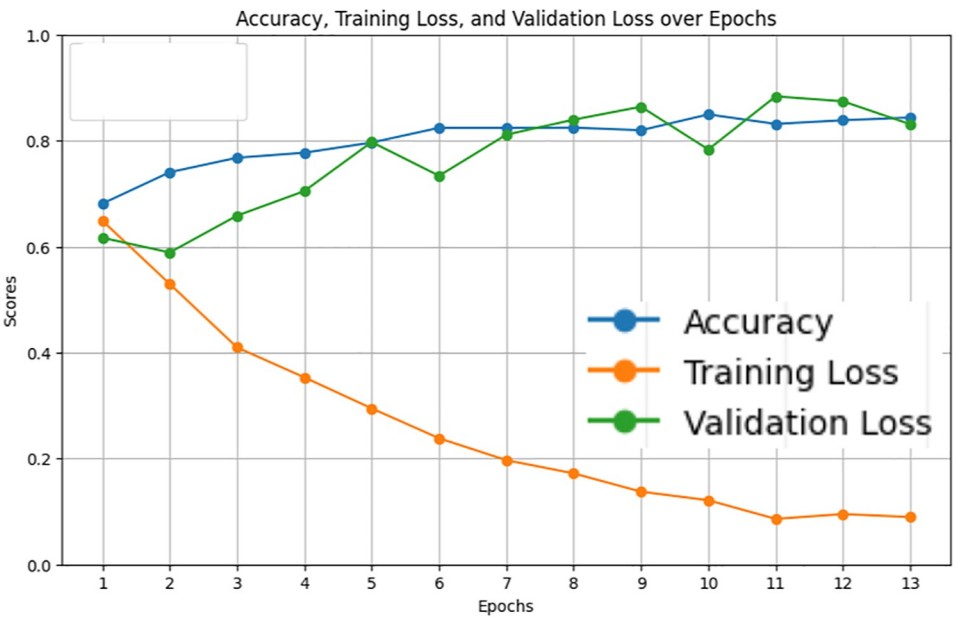

**Fig 12. Accuracy, training loss, and validation loss over Epochs for LastBert model.**

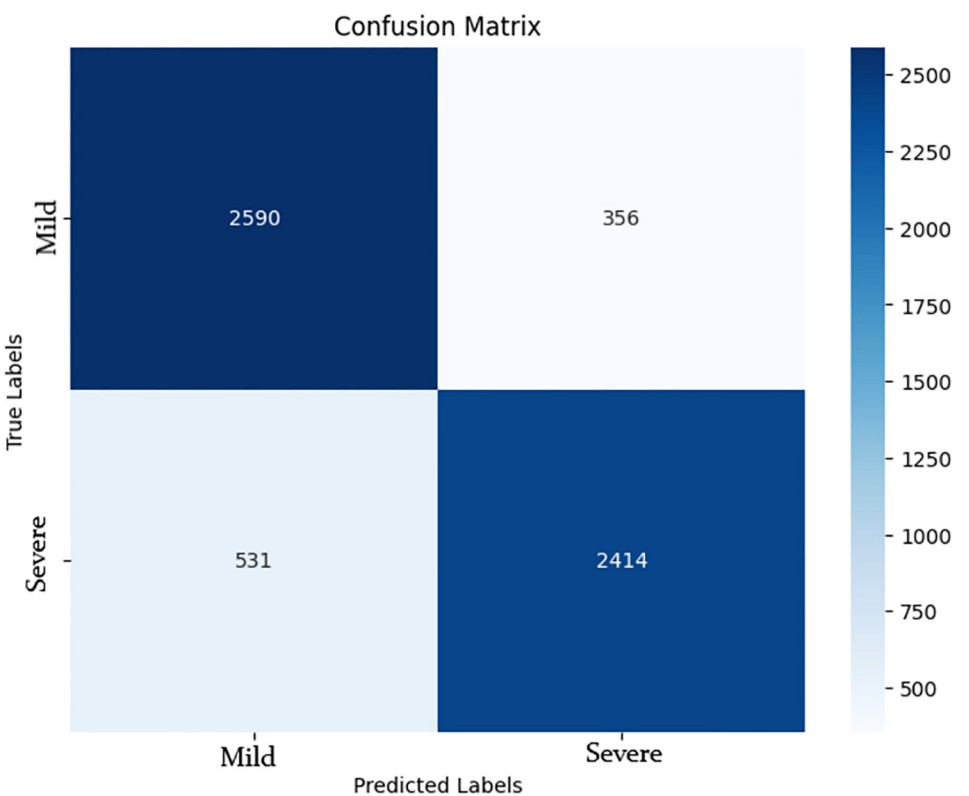

**Fig 13. Confusion matrix for LastBERT model.**

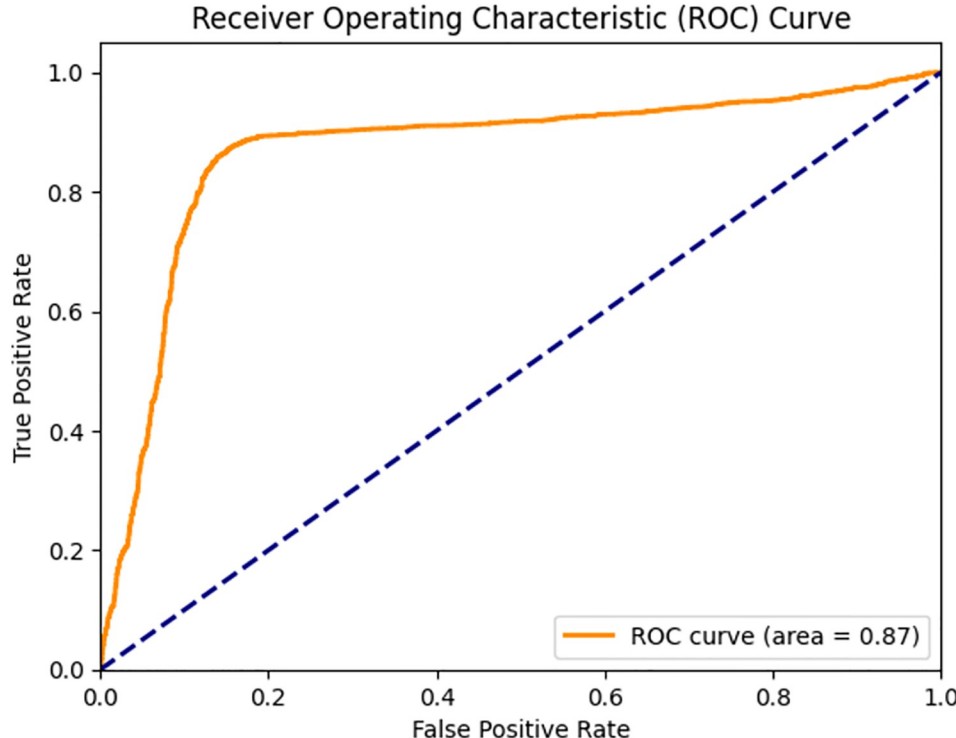

**Fig 14. ROC curve for LastBERT model.**

positive rate (1-specificity) across various threshold settings. The ability of the model to distinguish between classes is measured in the area under the ROC curve (AUROC). With an AUROC of 0.87, the model shows high discriminative ability, meaning it is good at separating Mild from Severe ADHD-related severity levels. The nearness of the ROC curve to the top left corner emphasizes even more the strong performance of the model.

**4.3.2 DistilBERT model.** The DistilBERT model exhibited the highest performance among the evaluated models for classifying ADHD-related severity concerns levels. The model was trained for 2 hours, 1 minute, and 1 second for 10 epochs. In which the model produced results of 87% accuracy, 87% f1 score, 87% precision, and 87% recall. Fig 15 presents the precision, recall, and F1 score over epochs, showing consistent and superior performance throughout the training process. The accuracy, training loss, and validation loss graphs (Fig 16) further emphasize the model's stability and convergence, with the validation loss showing minimal fluctuations.

The confusion matrix for the DistilBERT model (Fig 17) shows an overall accuracy of 87%. Specifically, the matrix indicates that out of 2946 Mild class instances, 2700 were correctly classified, and 246 were misclassified as Severe. For the Severe class, out of 2945 instances, 2400 were correctly classified, and 545 were misclassified as Mild. This balanced performance demonstrates the model's robustness in identifying both Mild and Severe cases effectively.

**4.3.3 ClinicalBERT model.** The ClinicalBERT model performed admirably in classifying ADHD-related severity concerns levels, with an overall accuracy of 86% with a training period of 2 hours, 27 minutes, and 45 seconds for 8 epochs. (Figs 18 and 19) display the precision,

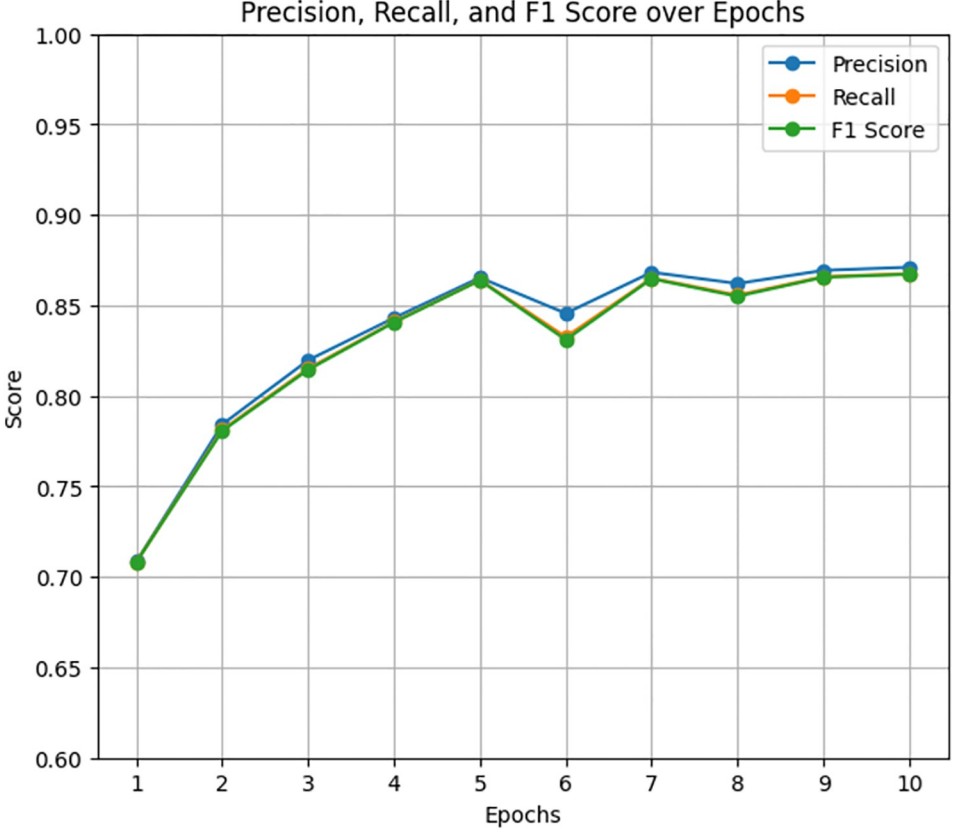

**Fig 15. Precision, recall, and F1 score over Epochs for DistilBERT model.**

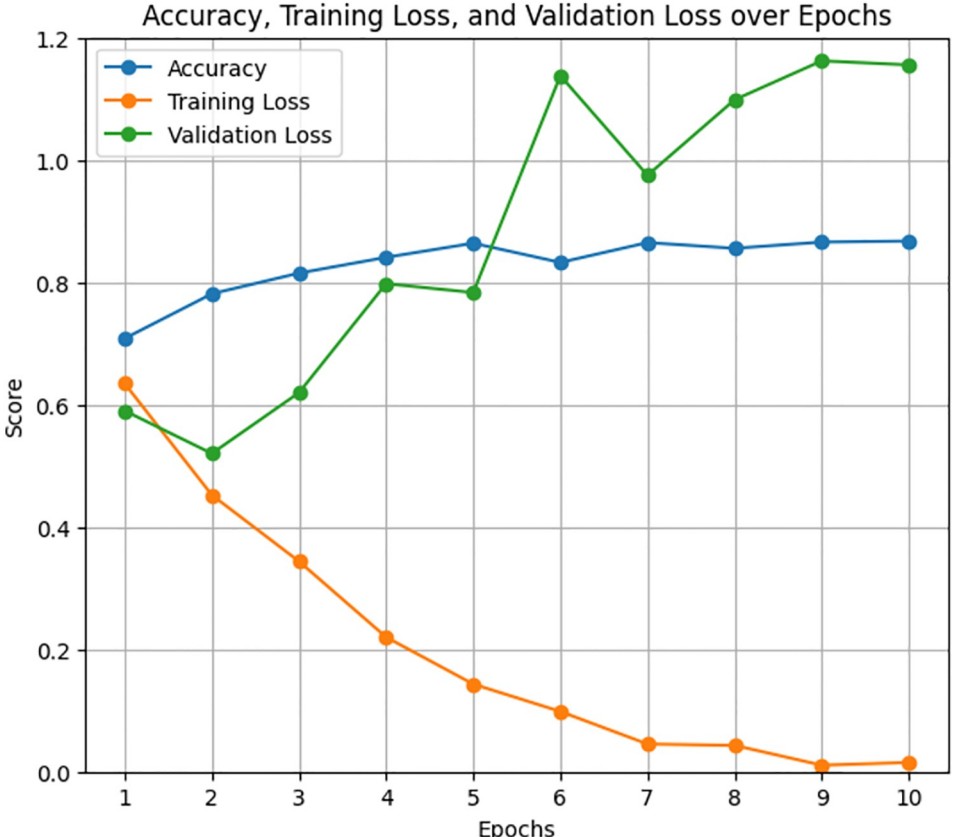

**Fig 16. Accuracy, training loss, and validation loss over Epochs for DistilBERT model.**

recall, F1 score, accuracy, training loss, and validation loss over epochs. The ClinicalBERT model shows stable performance with minor fluctuations in validation loss, indicating effective learning and generalization.

The confusion matrix for the ClinicalBERT model (Fig 20) indicates a balanced classification performance. Out of 2946 Mild class instances, 2676 were correctly classified, and 270 were misclassified as Severe. For the Severe class, out of 2945 instances, 2398 were correctly classified, and 547 were misclassified as Mild. This balanced classification accuracy reflects the model's robustness in distinguishing between the two classes.

**4.3.4 Overall comparison and insights.** Table 4 summarizes the overall performance of the three models in terms of accuracy, F1 score, precision, and recall. DistilBERT demonstrated the highest performance across all metrics, followed by ClinicalBERT and the LastBERT model. The detailed comparison of macro average and weighted average metrics is provided in Table 5.

With regard to the accuracy, precision, recall, and F1 score, the DistilBERT model showed overall better performance than all the others tested. DistilBERT specifically obtained an accuracy of 0.87, precision, recall, and F1 score all at 0.87. ClinicalBERT followed closely with 0.86 for precision, recall, and F1 scores all around, followed closely with an accuracy of 86%. With just 29 million parameters, our LastBERT model obtained an accuracy of 85%, precision, recall, and F1 scores all at 0.85. DistilBERT's 66 million parameters and ClinicalBERT's 110 million parameters contrast this.

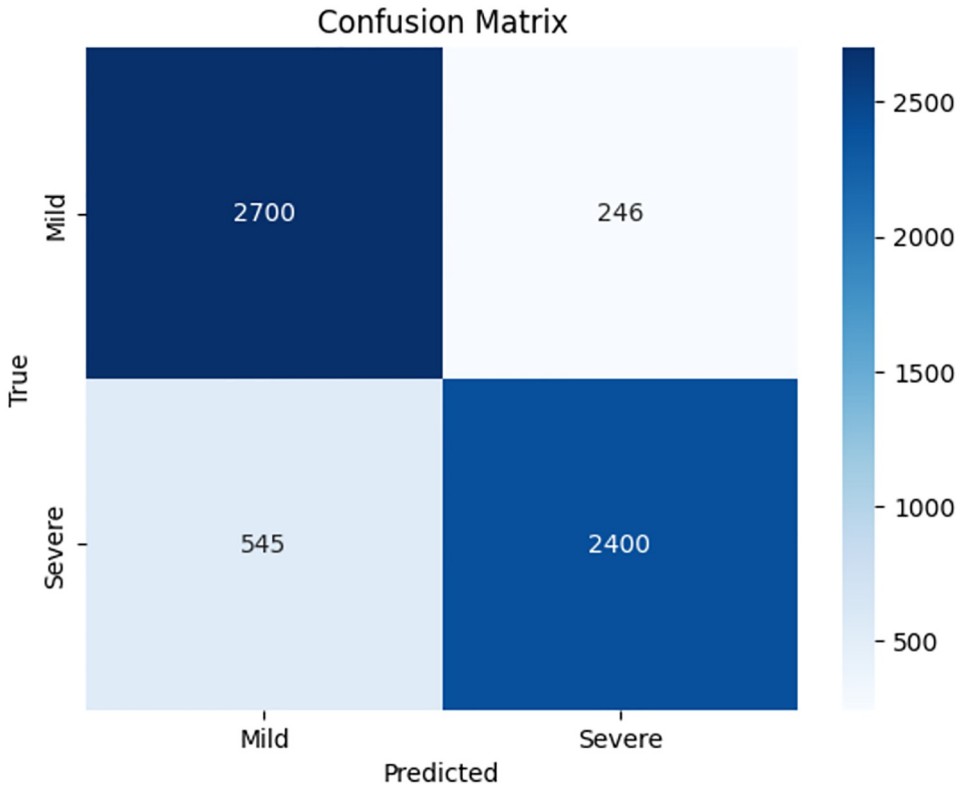

**Fig 17. Confusion matrix for DistilBERT model.**

The LastBERT model's parameter efficiency emphasizes the key variation between the models since it guarantees competitive performance with much fewer parameters. Furthermore, it has been noted that, in a smaller training time than the large models, equivalent outcomes can be obtained. The performance of the LastBERT model shows how well the information distillation technique generates smaller, effective models free from significant accuracy or other performance metric loss. These findings highlight the capacity of our simplified LastBERT model to efficiently categorize ADHD severity levels, therefore demonstrating its competitiveness with more established models in useful applications such as the categorization of mental health disorders from social media data. The results show the advantages of knowledge distillation in creating small, high-performance models perfect for use in resource-constrained conditions, therefore providing a well-balanced compromise between model size and performance.

**4.3.5 Comparisons of the results against the relevant works.**   Table 6 shows a comparison of several studies on ADHD text classification together with the models applied, datasets, and corresponding accuracy. With several BERT-based models applied to Reddit postings, our study shows better performance attaining accuracies of 85%, 87%, and 86% with LastBERT, DistilBERT, and ClinicalBERT, respectively. These findings highlight how strong advanced transformer models are in ADHD text categorization vs. conventional machine learning and NLP-based approaches.

**4.3.6 Comparison of knowledge distillation models on GLUE datasets.**   Table 7 presents an updated comparison of LastBERT with other knowledge distillation models such as Distil-BERT, MobileBERT, and TinyBERT, as well as BERT Base and BERT Large models, on various

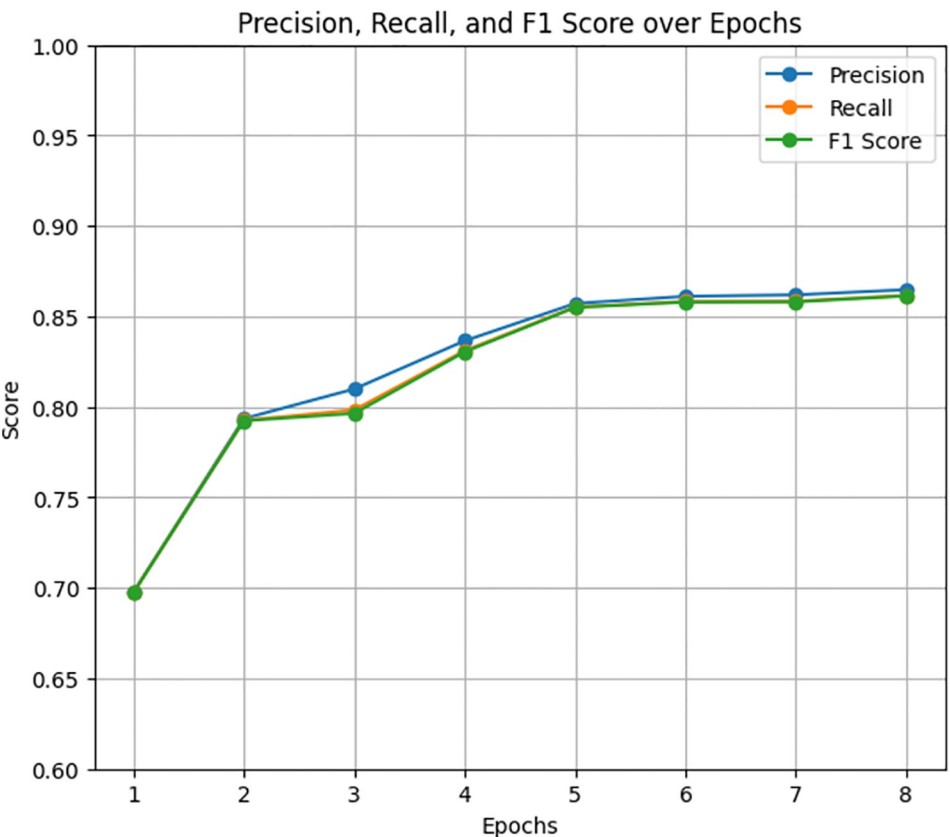

**Fig 18. Precision, recall, and F1 score over Epochs for ClinicalBERT model.**

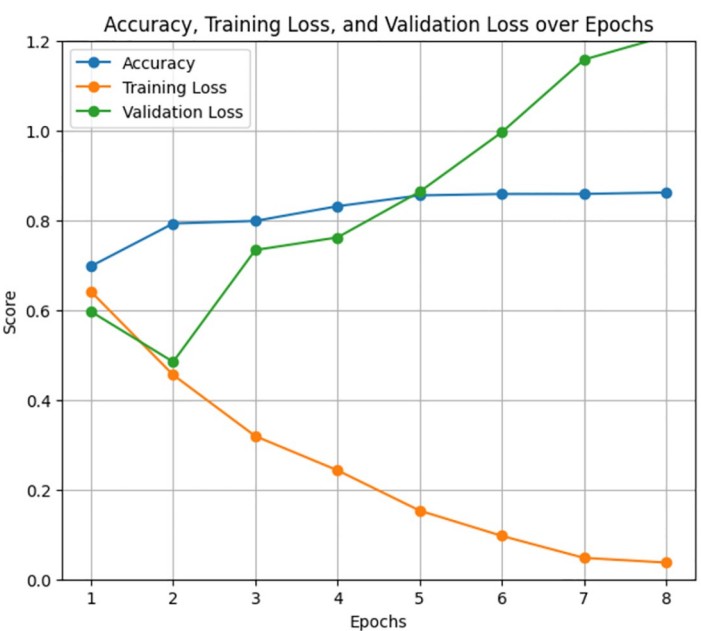

**Fig 19. Accuracy, training loss, and validation loss over Epochs for ClinicalBERT model.**

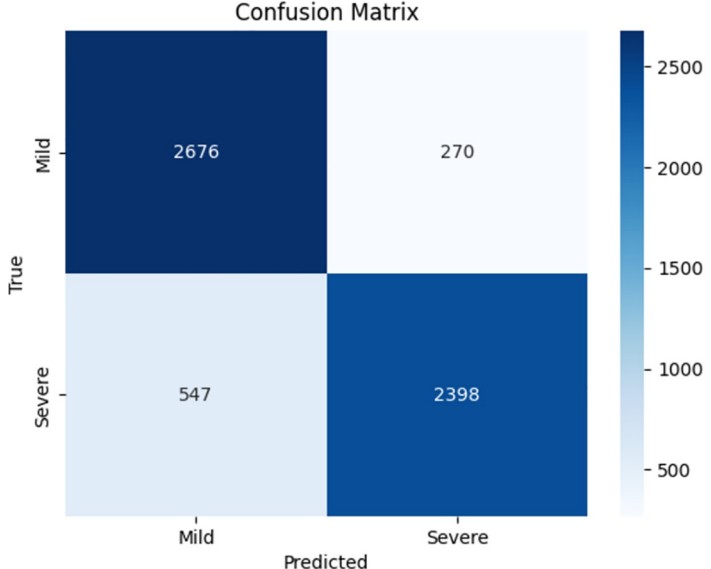

**Fig 20. Confusion matrix for ClinicalBERT model.**

GLUE datasets. Despite having fewer parameters than DistilBERT (66M vs. 29M), LastBERT achieves competitive performance across several tasks.

On the QQP dataset, LastBERT attains an F1 score of 0.77 and an accuracy of 82%, surpassing MobileBERT's F1 score of 0.70 and performing similarly to BERT Base's F1 score of 0.71. BERT Large performs slightly better with an F1 score of 0.72, highlighting the advantages of larger models in paraphrase identification.

For the MRPC dataset, LastBERT achieves an F1 score of 0.81 and an accuracy of 71%, trailing MobileBERT's F1 score of 0.87 (accuracy: 88%) but performing comparably to BERT Base (F1: 0.88, accuracy: 88%) and BERT Large (F1: 0.89, accuracy: 89%). TinyBERT also performs well on this dataset, matching MobileBERT's F1 score and accuracy.

On the SST-2 dataset, LastBERT records an accuracy of 82% and an F1 score of 0.82, trailing MobileBERT's 92% accuracy and DistilBERT's 91%. BERT Base and TinyBERT perform similarly, with 93% accuracy, while BERT Large leads with 94%. These results show that while LastBERT performs well in sentiment analysis, larger models like BERT Large achieve superior results.

On the CoLA dataset, LastBERT underperforms with a Matthews correlation coefficient of 0.17, compared to DistilBERT's 0.51 and MobileBERT's 0.50. BERT Base performs slightly better with a score of 0.52, while BERT Large achieves a 0.60 correlation. TinyBERT also matches DistilBERT with a 0.51 correlation. This discrepancy can be attributed to LastBERT's training

**Table 4. Performance comparison of LastBERT, DistilBERT, and ClinicalBERT on ADHD dataset.**

| Model | Accuracy | F1 Score | Precision | Recall | Parameters |
|---|---|---|---|---|---|
| **LastBERT (Our Model)** | 85% | 0.85 | 0.85 | 0.85 | 29M |
| **DistilBERT** | 87% | 0.87 | 0.87 | 0.87 | 66M |
| **ClinicalBERT** | 86% | 0.86 | 0.86 | 0.86 | 110M |

Performance comparison of LastBERT, DistilBERT, and ClinicalBERT on ADHD dataset, highlighting accuracy, F1 score, precision, recall, and number of parameters.

**Table 5. Weighted average comparison of LastBERT, DistilBERT, and ClinicalBERT on ADHD dataset.**

| Model | F1 Score | Precision | Recall |
|---|---|---|---|
| **LastBERT (Our Model)** | 0.85 | 0.85 | 0.85 |
| **DistilBERT** | 0.87 | 0.87 | 0.87 |
| **ClinicalBERT** | 0.86 | 0.86 | 0.86 |

Weighted average comparison of LastBERT, DistilBERT, and ClinicalBERT on the ADHD dataset, highlighting their performance in terms of F1 score, precision, and recall.

on the smaller WikiText-2 dataset (2M words), which lacks the linguistic structures required for such tasks.

For the STS-B dataset, LastBERT records a Spearman correlation of 0.35, significantly lower than DistilBERT's 0.87 and MobileBERT's 0.84. BERT Base and BERT Large lead with Spearman scores of 0.85 and 0.86, respectively. TinyBERT performs slightly lower at 0.83. These results highlight the importance of larger corpora and more sophisticated distillation techniques for tasks that require deep semantic understanding.

**Table 6. Comparison of related works and our study on ADHD text classification.**

| Study | Model | Method | Dataset | Accuracy | F1 |
|---|---|---|---|---|---|
| Malvika et al. [21] | BioClinical-BERT | NLP techniques & Fine Tuning | EHR Clinical | - | 0.78 |
| Peng et al. [22] | 3D CNN | Multimodal Neural Networks | ADHD-200 | 72.89% | - |
| Chen et al. [23] | Decision Tree | Machine Learning | Clinical Dataset | 75.03% | - |
| Alsharif et al. [24] | Random Forests | Machine Learning and Deep Learning | Reddit ADHD Dataset | 81% | - |
| Cafiero et al. [25] | Support Vector Classifier | Machine Learning | Self Defining Memories | - | 0.77 |
| Lee et al. [26] | RoBERTa | NLP Techniques & Fine Tuning | Reddit Mental Health | 76% | - |
| **Our Study** | **LastBERT (Our Model)** | Knowledge Distillation and NLP Techniques | Reddit Mental Health | **85%** | **0.85** |
| **Our Study** | **DistilBERT** | **NLP Techniques & Fine Tuning** | Reddit Mental Health | **87%** | **0.87** |
| Our Study | ClinicalBERT | NLP Techniques & Fine Tuning | **Reddit Mental Health** | **86%** | **0.86** |

This table compares ADHD text classification studies, focusing on models, methods, datasets, and key metrics. Some studies report F1-score instead of accuracy ("-" denotes unreported values). Our study applies NLP models (LastBERT, DistilBERT, ClinicalBERT) to the Reddit Mental Health dataset, demonstrating the potential of NLP in mental health diagnostics.

**Table 7. Performance comparison of LastBERT with Other BERT and knowledge distillation models on GLUE datasets.**

| Model | Parameters | MRPC | SST-2 | CoLA | QQP | MNLI | STS-B |
|---|---|---|---|---|---|---|---|
| **BERT Large [5]** | 340M | **0.89** | **94%** | **0.60** | 0.72 | **86%** | 0.86 |
| **BERT Base [5]** | 110M | 0.88 | 93% | 0.52 | 0.71 | 84% | 0.85 |
| **TinyBERT [7]** | 67M | 0.87 | 93% | 0.51 | 0.71 | 84% | 0.83 |
| **DistilBERT [6]** | 66M | 0.87 | 91% | 0.51 | 0.70 | 82% | **0.87** |
| **MobileBERT [47]** | 25.3M | 0.87 | 92% | 0.50 | 0.70 | 82% | 0.84 |
| **LastBERT (Our Model)** | 29M | 0.81 | 82% | 0.17 | **0.77** | 70% | 0.35 |

For MRPC and QQP datasets, the metric shown is the F1 score, which is a better indicator for tasks involving classification of paraphrases. For SST-2 and MNLI datasets, accuracy is used to measure the model's performance in sentiment analysis and natural language inference. For CoLA, the Matthews correlation coefficient is shown, which reflects grammatical acceptability. For STS-B, the Spearman correlation is provided to assess sentence similarity.

Despite these limitations, LastBERT's strong performance on tasks like QQP, MRPC, and SST-2 demonstrates its effectiveness in key NLP tasks s uch as paraphrase identification, sentiment analysis, and text classification. It offers a viable solution for resource-constrained environments, balancing performance with computational efficiency.

## 5 Discussion

This research highlights the effectiveness of knowledge distillation in developing lightweight NLP models. Our custom model, LastBERT, with only 29 million parameters, achieved solid performance on ADHD severity classification, with 85% accuracy, precision, recall, and F1 score. This demonstrates its suitability for resource-constrained environments, providing a viable alternative to larger models like DistilBERT (87%) and ClinicalBERT (86%). Despite this success, trade-offs emerged, as some information was inevitably lost during distillation, leading to slightly lower performance compared to the larger models. The reliance on Reddit mental health posts, though useful for demonstrating the model's potential, introduces potential bias as these posts may not represent broader clinical or demographic realities. Further validation with more diverse datasets, including clinical records, will ensure better generalization. BERT-based models were chosen for the knowledge distillation approach due to their proven efficiency across various NLP tasks, such as text classification, sentiment analysis, and question answering. BERT strikes an optimal balance between performance and computational feasibility, making it suitable for knowledge distillation. In contrast, large language models (LLMs) like GPT and LLaMA, while powerful, are often impractical for training or fine-tuning on platforms like Colab and Kaggle due to their large size and limited resources. A deeper analysis reveals that our model, LastBERT performed consistently across sentiment analysis, question answering, and text classification tasks. On the QQP dataset, LastBERT achieves an 82% accuracy and an F1 score of 0.77, outperforming MobileBERT's 70% accuracy, demonstrating its superior ability to identify paraphrases effectively. However, it did struggle with more nuanced NLP datasets, such as CoLA (Corpus of Linguistic Acceptability) and STS-B (Semantic Textual Similarity Benchmark). We attribute this to the use of WikiText-2 during distillation, which lacks the linguistic structures and semantic elements relevant to these tasks. While WikiText-2 offers general language modeling benefits, it falls short of capturing the syntactic nuances required for CoLA and the fine-grained semantic similarities necessary for STS-B. Compared to DistilBERT and MobileBERT, these models were trained using the BooksCorpus (1 billion words) and English Wikipedia (2.5 billion words) datasets, establishing their excellent results. Due to computational constraints, Wikitext-2 (2M words) was opted as a feasible dataset for our study. Future studies could explore using a more specialized corpus, aligned with the linguistic characteristics of CoLA or STS-B, to enhance the model's performance on these benchmarks.

In light of recent studies demonstrating the effectiveness of fine-tuning pre-trained models on small datasets [18–20], this work also explores the trade-offs between fine-tuning and knowledge distillation. While fine-tuning allows models to adapt to task-specific nuances and has shown remarkable results in various NLP tasks, it requires substantial computational resources. One notable advantage of the LastBERT model is its efficiency in terms of both memory usage and inference time. Due to its smaller size and fewer parameters, LastBERT requires significantly less memory compared to fine-tuned BERT-based models like BERT Base or BERT Large. This makes it particularly suitable for deployment in environments where computational resources are limited, such as mobile devices or real-time applications. Additionally, LastBERT's faster inference time makes it ideal for scenarios where quick response times are crucial, further enhancing its practicality in resource-constrained settings.

Knowledge distillation, in contrast, creates more efficient models by sacrificing some level of task-specific optimization but yields models that are more suitable for real-time applications and environments with limited computational resources. This balance between efficiency and performance is critical, and both approaches have their merits depending on the deployment context.

Future research could explore transfer learning from other mental health tasks to strengthen the model's adaptability. Applying ensemble approaches by integrating LastBERT with other lightweight models might further enhance performance. Moreover, optimizing the model for edge deployment would enable real-time monitoring and intervention in resource-limited settings, extending its impact beyond academic research.

In summary, this study demonstrates that knowledge distillation offers a promising path to building compact, high-performing NLP models. While LastBERT achieves a compelling balance between efficiency and effectiveness, addressing these limitations will be critical to unlocking its full potential for both clinical and non-clinical applications.

## 6 Conclusion

This study answers the question posed by its title by demonstrating that while larger models generally yield better results, smaller models like LastBERT can achieve competitive performance with significant efficiency. With only 29 million parameters, LastBERT reached 85% accuracy, precision, recall, and F1 score in ADHD severity classification, proving that smaller models can approach the effectiveness of their larger counterparts. This makes them viable alternatives, especially in resource-constrained environments. Accurate ADHD severity classification plays a critical role in mental health diagnostics by enabling timely interventions and enhancing care prioritization. However, LastBERT performs within 1%–2% of larger models like BERT Base and MobileBERT on paraphrase identification tasks (QQP, MRPC), demonstrating strong performance despite its smaller size. However, it lags behind by around 8%–10% on sentiment analysis tasks (SST-2) and performs about 50%–55% worse on more complex tasks like CoLA and STS-B, which require deeper linguistic and semantic understanding, reflecting the importance of choosing the right pretraining corpus. Future work should explore domain-specific corpora and interpretability methods to build trust in clinical settings. Ultimately, this research shows that knowledge-distilled models strike a practical balance between performance and accessibility, demonstrating that smaller models can closely rival larger ones without requiring extensive computational resources.

## 7 Dataset, model, and code availability

The Dataset used in this study is publicly available. The Model is implemented using Last-BERT. The Code for knowledge distillation and classification is accessible in the repository.

## Supporting information

**S1 File.**
(CSV)

## Acknowledgments

Thanks to Dr. Ashrafuzzaman Khan Sir for his guidance and knowledge that brought me into the realm of AI and Natural Language Processing.

## Author Contributions

**Conceptualization:** Ahmed Akib Jawad Karim.

**Data curation:** Ahmed Akib Jawad Karim.

**Formal analysis:** Ahmed Akib Jawad Karim, Kazi Hafiz Md. Asad.

**Funding acquisition:** Md. Golam Rabiul Alam.

**Investigation:** Ahmed Akib Jawad Karim, Md. Golam Rabiul Alam.

**Methodology:** Ahmed Akib Jawad Karim.

**Project administration:** Ahmed Akib Jawad Karim.

**Resources:** Ahmed Akib Jawad Karim.

**Software:** Ahmed Akib Jawad Karim.

**Supervision:** Ahmed Akib Jawad Karim, Md. Golam Rabiul Alam.

**Validation:** Ahmed Akib Jawad Karim, Kazi Hafiz Md. Asad.

**Visualization:** Kazi Hafiz Md. Asad, Md. Golam Rabiul Alam.

**Writing – original draft:** Ahmed Akib Jawad Karim.

**Writing – review & editing:** Ahmed Akib Jawad Karim, Md. Golam Rabiul Alam.

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
