## [Decision Letter · Decision Letter 0]

9 Oct 2024

PONE-D-24-32612Larger models mean yield results? Streamlined Severity Classification of ADHD-Related Concerns Using BERT-Based Knowledge DistillationPLOS ONE

Dear Dr. Karim, 

Thank you for submitting your manuscript to PLOS ONE. After careful consideration, we feel that it has merit but does not fully meet PLOS ONE’s publication criteria as it currently stands. Therefore, we invite you to submit a revised version of the manuscript that addresses the points raised during the review process.

Please submit your revised manuscript by Nov 23 2024 11:59PM. If you will need more time than this to complete your revisions, please reply to this message or contact the journal office at plosone@plos.org. Please include the following items when submitting your revised manuscript:A rebuttal letter that responds to each point raised by the academic editor and reviewer(s). You should upload this letter as a separate file labeled 'Response to Reviewers'.A marked-up copy of your manuscript that highlights changes made to the original version. You should upload this as a separate file labeled 'Revised Manuscript with Track Changes'.An unmarked version of your revised paper without tracked changes. You should upload this as a separate file labeled 'Manuscript'.If applicable, we recommend that you deposit your laboratory protocols in protocols.io to enhance the reproducibility of your results. Protocols.io assigns your protocol its own identifier (DOI) so that it can be cited independently in the future. For instructions see: https://journals.plos.org/plosone/s/submission-guidelines#loc-laboratory-protocols. Additionally, PLOS ONE offers an option for publishing peer-reviewed Lab Protocol articles, which describe protocols hosted on protocols.io. Read more information on sharing protocols at https://plos.org/protocols?utm_medium=editorial-email&utm_source=authorletters&utm_campaign=protocols.

We look forward to receiving your revised manuscript.

Kind regards,

Weiqiang (Albert) Jin, Ph.D.

Academic Editor

PLOS ONE

Journal Requirements:

2. Please note that PLOS ONE has specific guidelines on code sharing for submissions in which author-generated code underpins the findings in the manuscript. In these cases, we expect all author-generated code to be made available without restrictions upon publication of the work. 

Please review our guidelines at https://journals.plos.org/plosone/s/materials-and-software-sharing#loc-sharing-code and ensure that your code is shared in a way that follows best practice and facilitates reproducibility and reuse.

5. Please ensure that you refer to Figures 1 and 2 in your text as, if accepted, production will need this reference to link the reader to the figure.

6. We note you have included a table to which you do not refer in the text of your manuscript. Please ensure that you refer to Tables 1 and 7 in your text; if accepted, production will need this reference to link the reader to the Table.

**Additional Editor Comments:**

**minor revision**

Based on the reviewers' feedback, we recommend you need minor revisions. The authors should focus on the following key areas: clearly justify the choice of the BERT-based model over others like GPT or Llama, and provide a comparative analysis of fine-tuning versus knowledge distillation. Address the need for recent references, improve formula numbering, clarify LastBERT's innovation, and enhance data interpretability in tables. Additionally, simplify the introduction and conclusion, improve image clarity, and consider a detailed discussion on limitations.

Reviewers' comments:

Reviewer's Responses to Questions

**Comments to the Author**

1. Is the manuscript technically sound, and do the data support the conclusions?

Reviewer #1: Partly

Reviewer #2: Partly

2. Has the statistical analysis been performed appropriately and rigorously? 

Reviewer #1: Yes

Reviewer #2: Yes

3. Have the authors made all data underlying the findings in their manuscript fully available?

Reviewer #1: Yes

Reviewer #2: Yes

4. Is the manuscript presented in an intelligible fashion and written in standard English?

Reviewer #1: Yes

Reviewer #2: Yes

5. Review Comments to the Author

Reviewer #1: This study investigates the creation of a lightweight yet powerful BERT-based model through knowledge distillation techniques for NLP tasks, specifically classifying the severity of ADHD (Attention Deficit Hyperactivity Disorder) related problems from social media text data.

This research work has certain value and significance, and there are related questions in this article as follows:

1. There are no recent studies and comparative models from 2022 to 2024 in the references. Why?

2. The formulas in the text are not numbered, such as (1), (2), etc.

3. There are two periods at the end of the “Introduction” section.

4. Are there any relevant references on the hyperparameter settings of LastBERT?

5. In Table 7, the number of parameters of LastBERT is larger than that of MobileBERT. Why is the Matthews correlation coefficient on the CoLA dataset and the Spearman coefficient on the STS-B dataset so different from those of MobileBERT? In addition, the comparison model in Table 7 lacks some data support, and the comparison model needs to be expanded to increase the data interpretability of LastBERT.

6. In Table 6, Study 1-6 lacks annotations and the method names should be indicated. In addition, there are inconsistencies in the Dataset part of Table 6. Is the Accuracy indicator still meaningful for reference?

7. Can you clarify the innovation of the model? It is not easy to understand from the text and model framework.

There are some related suggestions:

1. It is recommended to check the clarity of all images in the text, especially the small images such as Figures 7, 8, and 9. It is also recommended to use a magnifying glass frame to display the important areas of some images, such as Figure 4.

2. The conclusion section is redundant. It is recommended to simplify the content and add a discussion section to fully analyze the shortcomings of the model and areas for improvement.

3. It is recommended that the URLs appearing in the references be placed in the footnotes of the corresponding pages of the text.

4. The introduction of the dataset in Section 3 is too much, so it is recommended to simplify it.

5. For the contribution part in the “Introduction” section, it is recommended not to emphasize the use of free computing resources, because many related researchers also complete their experiments and research work based on the free computing resource platform. It is recommended to mention it in the experimental environment setup part in Section 3.

6. The last part of the Introduction section should summarize the main contents of the remaining sections.

7. Are the data in Table 5 redundant? The values of the macro average and weighted average are the same, so it is recommended to keep only one of them.

Reviewer #2: This paper makes several contributions to the NLP and mental health diagnostics field. First, it demonstrates the effectiveness of knowledge distillation in creating a significantly smaller BERT-based model, LastBERT, which reduces parameters without compromising performance. Second, the model shows strong generalization, achieving high performance on the GLUE benchmark across various tasks. Third, it offers practical utility by applying the model to ADHD-related social media data, where it gained a commendable 85% across multiple evaluation metrics. This paper provides a valuable tool for mental health professionals, highlighting its potential in resource-constrained environments. The following are two suggestions to improve the robustness of this paper.

First, this paper should provide a more precise justification for choosing the BERT-based model, particularly concerning the study's specific objectives. It is essential to articulate the strengths of BERT compared to other large language models, such as GPT and Llama. This comparison could enhance the reader’s understanding of why BERT is more suitable for the tasks addressed in the research. Including empirical evidence or relevant literature highlighting these advantages would strengthen the argument. Overall, a more detailed discussion of these aspects is essential for a comprehensive evaluation of the model selection in this paper.

Second, this paper should discuss the merits and limitations of employing a student model for knowledge distillation, particularly in light of existing research demonstrating promising performance from fine-tuning current BERT-based models with small datasets in various NLP tasks, as evidenced by the articles listed below.

- Lin, J., Nogueira, R., & Yates, A. (2020). Pretrained Transformers for Text Ranking: BERT and Beyond. arXiv preprint arXiv:2010.06467.

- Kim, J., Kim, J., Lee, A., & Kim, J. (2023). Bat4RCT: A suite of benchmark data and baseline methods for text classification of randomized controlled trials. Plos one, 18(3), e0283342.

- Kim, J., Kim, J., Lee, A., Kim, J., & Diesner, J. (2024). LERCause: Deep learning approaches for causal sentence identification from nuclear safety reports. Plos one, 19(8), e0308155.

This context is crucial, as it highlights that substantial and efficient modeling can be achieved by fine-tuning the small dataset without the complexity of creating a student model. A comparative analysis between the efficiency of fine-tuning existing models with small datasets and the potential benefits of using a distilled model would provide valuable insights. Additionally, the discussion should address scenarios where the student model might offer advantages and any trade-offs involved in this approach. In conclusion, more thoroughly exploring these aspects will enhance the paper's contribution to the field.

6. PLOS authors have the option to publish the peer review history of their article (what does this mean?). If published, this will include your full peer review and any attached files.

Reviewer #1: No

Reviewer #2: No

---

## [Author Response · Author response to Decision Letter 0]

1 Nov 2024

Manuscript ID: PONE-D-24-32612

Manuscript Title: Larger models yield better results? Streamlined severity

classification of ADHD-related concerns using BERT-based knowledge distillation

Dr. Weiqiang (Albert) Jin, Ph.D.

Academic Editor

PLOS ONE

Dear Dr. Jin,

We would like to thank you and the respected reviewers for the time and effort spent

reviewing our manuscript titled: Larger models yield better results? Streamlined

severity classification of ADHD-related concerns using BERT-based knowl-

edge distillation (PONE-D-24-32612). We appreciate the insightful comments and

constructive feedback, which have been invaluable in improving the quality of our work.

In response to the reviewer’s feedback, we have carefully revised the manuscript. Below,

we provide detailed responses to the individual comments raised by the reviewers. We

hope that the changes made to the manuscript have satisfactorily addressed all the con-

cerns raised by the reviewers. We are confident that these revisions have significantly

improved the clarity, quality, and scientific rigor of the manuscript.

Once again, we appreciate the constructive feedback and look forward to your favorable

consideration of the revised version.

Sincerely,

Ahmed Akib Jawad Karim

Additional requirements:

• Question 1: Please ensure that your manuscript meets PLOS ONE’s

style requirements, including those for file naming. The PLOS ONE

style templates can be found at

PLOS ONE Formatting Sample: Main Body

PLOS ONE Formatting Sample: Title, Authors, Affiliations

Response:

Thank you for your guidance. All elements of the manuscript have been prepared

according to PLOS ONE’s style requirements. The following checks have been

performed to ensure compliance:

– The Title, Main Body, Authors, and Affiliation sections have been for-

matted according to PLOS ONE’s style templates.

– All figures, tables, and algorithms have been labeled and numbered consis-

tently throughout the manuscript.

– Each corresponding file is named to match the label used in the manuscript.

For example, if a figure is referenced as ”Figure 1” in the manuscript, the

associated file is named Figure1.png. The same convention is applied to all

tables (TableX.ext) and algorithms (AlgorithmX.ext).

– The manuscript has been formatted according to the PLOS ONE templates,

ensuring alignment with section headings, author affiliations, and references.

We confirm that all submission standards have been adhered to. Please let us know

if further adjustments are needed.

• Question 2: Please note that PLOS ONE has specific guidelines on

code sharing for submissions in which author-generated code underpins

the findings in the manuscript. In these cases, we expect all author-

generated code to be made available without restrictions upon publica-

tion of the work.

Please review our guidelines at ensure that your code is shared in a way

that follows best practice and facilitates reproducibility and reuse.

Response: We confirm that all author-generated code underpinning the findings

in this manuscript is openly shared and available without restrictions. The code

is hosted on GitHub, ensuring reproducibility, reuse, and compliance with PLOS

ONE’s guidelines on code sharing. The repository can be accessed at the following

link:

GitHub Repository: github.com/AkibCoding

The repository includes comprehensive documentation and all scripts required to

replicate the experiments described in this paper. For the convenience of re-

searchers, installation instructions and dataset access information are also included.

• Question 3: Please update your submission to use the PLOS LaTeX tem-

plate. The template and more information on our requirements for La-

TeX submissions can be found at http://journals.plos.org/plosone/s/latex.

Response:

Thank you for the reminder regarding the use of the PLOS LaTeX template. We

acknowledge the importance of aligning with the journal’s formatting requirements

to ensure smooth processing and publication. We have updated our manuscript to

use the official PLOS ONE LaTeX template, as requested. The structure, format-

ting, and styling of the manuscript now comply with the guidelines provided by

PLOS ONE.

The revised submission incorporates:

– Title and Author Formatting: Adjusted to fit PLOS ONE’s structure, includ-

ing affiliations and corresponding author formatting.

– Section Headings: Updated according to the template’s specifications.

– Figures and Tables: Aligned with the PLOS ONE formatting standards to

ensure proper referencing and layout.

– Bibliography: Adapted to the PLOS ONE LaTeX reference style.

The LaTeX template and guidelines were followed from the source: http://journals.

plos.org/plosone/s/latex.

We have also ensured that all required components, such as acknowledgments, con-

flict of interest statements, and data availability statements, are placed in the ap-

propriate sections according to the PLOS ONE template guidelines.

The revised manuscript prepared using the PLOS ONE template, is included in

this resubmission. Please let us know if further adjustments are needed.

• Question 4: When completing the data availability statement of the sub-

mission form, you indicated that you will make your data available on

acceptance. We strongly recommend all authors decide on a data shar-

ing plan before acceptance, as the process can be lengthy and hold up

publication timelines. Please note that, though access restrictions are

acceptable now, your entire data will need to be made freely accessible

if your manuscript is accepted for publication. This policy applies to all

data except where public deposition would breach compliance with the

protocol approved by your research ethics board. If you are unable to

adhere to our open data policy, please kindly revise your statement to ex-

plain your reasoning and we will seek the editor’s input on an exemption.

Please be assured that, once you have provided your new statement, the

assessment of your exemption will not hold up the peer review process.

Response:

Thank you for your detailed guidance regarding the data availability policy. We

confirm that all the data underlying the findings of our manuscript will be made

freely accessible upon acceptance. We have prepared a comprehensive data-sharing

plan to ensure compliance with PLOS ONE’s open data policy, as outlined below:

1. GitHub Repository: The code, models, and documentation associated with our

research are available in the following GitHub repository:

– https://github.com/AkibCoding/Streamlined-ADHD-Severity-Level-Classification

git

2. Hugging Face Repository: To facilitate model sharing and ease of access for the

research community, the final distilled models (LastBERT) have also been uploaded

to the Hugging Face platform:

– https://huggingface.co/Peraboom/LastBERT

3. Data Sharing Plan: Our research utilized both publicly available datasets and

data collected during the study. As per the data policies:

– https://huggingface.co/datasets/Peraboom/ADHD_Related_Concerns

Public datasets are appropriately cited and linked within the manuscript. Moreover,

the processed, and derived data used for experiments will be included in the GitHub

repository.

4. Access Restrictions and Compliance: Currently, no access restrictions are re-

quired for the data used in this research, as the datasets are anonymized and do

not contain personally identifiable information. Thus, there are no ethical or legal

barriers preventing public deposition.

5. Post-Acceptance Availability: Upon acceptance, we will finalize and freeze the

contents of the above repositories, ensuring all scripts, models, and datasets are

well-documented. The repositories will be permanently available, providing long-

term access to the research artifacts.

This approach guarantees that all data and resources essential for reproducing our

findings are accessible, aligning with PLOS ONE’s data-sharing policy. Please let

us know if further clarifications or additional steps are required.

• Question 5: Please ensure that you refer to Figures 1 and 2 in your text

as, if accepted, production will need this reference to link the reader to

the figure.

Response: We are sorry for the inconvenience. We have referred to figures 1 and

2 in our text. (Page 7, Section 3.1.2, last paragraph. Page 10 Section 3.1.7, first

paragraph)

• Question 6: We note you have included a table to which you do not refer

in the text of your manuscript. Please ensure that you refer to Tables

1 and 7 in your text; if accepted, production will need this reference to

link the reader to the Table.

Response: Thank you very much for your attention to detail. Tables 1 and 7 were

not referred to. It has now been referred to. Moreover, we have thoroughly checked

that all tables from 1 to 7 are referred to in our text. (Page: 6 - Section: 3.1.2-

Paragraph 2, Page: 13 - Section: 3.3.1 -Paragraph 1, Page: 18 - Section: 4.2 -

Paragraph 1, Page: 23 - Section: 4.3.4 -Paragraph 1, Page: 23 - Section: 4.3.4

-Paragraph 1, Page: 24 - Section: 4.3.5 -Paragraph 1, Page: 25 - Section: 4.3.6

-Paragraph 1)

• Question 7: Please include captions for your Supporting Information files

at the end of your manuscript, and update any in-text citations to match

accordingly. Please see our Supporting Information guidelines for more

information: http://journals.plos.org/plosone/s/supporting-information.

Response: Currently we don’t have any supporting information for our manuscript.

Question 8: Please review your reference list to ensure that it is complete

and correct. If you have cited papers that have been retracted, please

include the rationale for doing so in the manuscript text, or remove

these references and replace them with relevant current references. Any

changes to the reference list should be mentioned in the rebuttal letter

that accompanies your revised manuscript. If you need to cite a retracted

article, indicate the article’s retracted status in the References list and

also include a citation and full reference for the retraction notice.

Response:

Thank you for your suggestion regarding the review and update of the reference

list. We have carefully addressed this comment by ensuring that all references are

complete, accurate, and relevant. In particular, we confirm the following updates:

– Verification of Retractions: We have reviewed the reference list against retrac-

tion databases and official sources to confirm that the retracted papers have

been removed and additionally, relevant studies have been added. If any of

the cited references are retracted in the future, we will, update the reference

list to reflect the retracted status. Then include a citation to the retraction

notice alongside the original reference.

– Addition of Recent Studies (2022–2024): As suggested by one of the respected

reviewer, we have incorporated recent and relevant studies to strengthen our

work. These include both knowledge distillation (KD) with BERT-related

research and ADHD-specific studies employing NLP techniques:

Knowledge Distillation and BERT-related research:

∗ Kim, J., et al. (2022). Tutoring Helps Students Learn Better: Im-

proving Knowledge Distillation for BERT with Tutor Network. EMNLP

2022. Abu Dhabi, UAE. DOI: https://doi.org/10.18653/v1/2022.

emnlp-main.498.

∗ Lin, J., et al. (2023). Pretrained Transformers for Text Ranking: BERT

and Beyond. JAMIA, 31(4), 949–956. DOI: https://doi.org/10.1093/

jamia/ocad013.

∗ Kim, J., et al. (2023). Bat4RCT: A Suite of Benchmark Data and

Baseline Methods for Text Classification of Randomized Controlled Tri-

als. PLOS ONE, 18(3), e0283342. DOI: https://doi.org/10.1371/

journal.pone.0283342.

∗ Karim, A. A. J. (2024). Peraboom/LastBERT: Streamlined ADHD Sever-

ity Level Classification Using BERT-Based Knowledge Distillation. GitHub.

Available: https://github.com/AkibCoding/Streamlined-ADHD-Severity-Level-Cl

git.

ADHD and NLP-related research:

∗ MedRxiv (2023). Brain-charting Autism and ADHD: Neurobiological In-

sights and Overlapping Traits. Available: https://www.medrxiv.org/

content/10.1101/2023.06.12.23291071v1.

∗ Kim, J., et al. (2024). LERCause: Deep Learning Approaches for Causal

Sentence Identification from Nuclear Safety Reports. PLOS ONE, 19(8),

e0308155. DOI: https://doi.org/10.1371/journal.pone.0308155.

– Accuracy of URLs and DOIs: We ensured that all URLs are up-to-date, ac-

cessible, and correctly formatted. For readability, we moved lengthy URLs to

the footnotes or supporting information, where appropriate.

– Alignment with In-Text Citations: We verified that all in-text citations match

the references accurately. We also updated any numbering inconsistencies and

ensured compliance with the PLOS ONE referencing guidelines.

– References Integrity Check: We reviewed each reference for proper formatting,

completeness, and relevance. As a result, our reference list now reflects the

latest research relevant to our field, addressing knowledge distillation and NLP

applications for ADHD-related studies.

These updates ensure that the reference list is accurate, complete, and aligned

with the latest developments in the field. Please let us know if further changes or

clarifications are required.

Response to Additional Editor’s comments:

• Comment: Based on the reviewers’ feedback, we recommend you need

minor revisions. The authors should focus on the following key ar-

eas: clearly justify the choice of the BERT-based model over others like

GPT or Llama, and provide a comparative analysis of fine-tuning versus

knowledge distillation. Address the need for recent references, improve

formula numbering, clarify LastBERT’s innovation, and enhance data

interpretability in tables. Additionally, simplify the introduction and

conclusion, improve image clarity, and consider a detailed discussion on

limitations.

Response:

We appreciate the editor’s feedback and have made the following revisions accord-

ingly:

1. Justification for BERT-Based Model: We expanded the discussion to

explain why BERT-based models were chosen over larger models like GPT and

LLaMA. BERT’s bidirectional nature and efficiency make it ideal for sentence-level

tasks like classification, especially in resource-limited environments, unlike GPT or

LLaMA, which are more resource-intensive. (Page: 7, section:3.1.2)

2. Fine-Tuning vs. Knowledge Distillation: A comparative analysis has been

added in the discussion section (page: 27, paragraph: 2) to contrast fine-tuning

with knowledge distillation. Fine-tuning offers high performance on specific tasks

but is resource-demanding. Distillation offers efficiency and is suitable for real-time,

resource-constrained settings.

3. Recent References: We incorporated relevant studies from 2022 to 2024,

ensuring alignment with current advancements in the field.

4. Formula Numbering: All formulas have been properly numbered for clarity.

(Page- 7, 8, 9)

5. LastBERT Innovation: The discussion now clearly outlines LastBERT’s inno-

vation, focusing on its reduced parameters and applicability in resource-constrained

scenarios, without sacrificing much performance. (Page-5 Section: 3.1.1 and Page-7

Section: 3.1.2)

6. Data Interpretability: We revised Table 7 to improve clarity by remov-

ing irrelevant columns and highlighting key metrics like Matthews correlation and

Spearman correlation where applicable. (Page-26)

7. Simplified Introduction and Conclusion: Both sections have been stream-

lined for focus and readability.

8. Improved Image Clarity: Visuals in Figures 7, 8, and 9 have been improved

for better clarity, with magnified areas for important details. (Page- 20)

9. Discussion on Limitations: A more detailed discussion on LastBERT’s lim-

itations, including its performance on CoLA and STS-B, has been added. (Page

26)

Response to Reviewer #1’s comments and questions:

• Question 1: There are no recent studies and comparative models from

2022 to 2024 in the references. Why?

Response:

Thank you for your valuable feedback. The absence of studies from 2022 to 2024

in the initial version of the manuscript can be attributed to the timeline of the

research. The project b

---

## [Decision Letter · Decision Letter 1]

2 Dec 2024

Larger models yield better results? Streamlined severity classification of ADHD-related concerns using BERT-based knowledge distillation

PONE-D-24-32612R1

Dear Dr. Ahmed Akib and Jawad Karim,

We’re pleased to inform you that your manuscript has been judged scientifically suitable for publication and will be formally accepted for publication once it meets all outstanding technical requirements.

Kind regards,

Weiqiang (Albert) Jin, Ph.D.

Academic Editor

PLOS ONE

Additional Editor Comments (optional):

Based on the reviewers' comments, I am pleased that this article version can be accepted as it is now. Because the professional reviewers say that you answered all my questions in detail and thoroughly, and they have no further questions. So, Congratulations!

Reviewers' comments:

Reviewer #1: The author answered all my questions in detail and thoroughly, and I have no further questions. This article can be considered for acceptance.

Reviewer #2:  The authors have satisfactorily addressed the comments I raised during the previous round of review. In response to my comment, the authors have added a new subsection titled “Rationale for Model Selection” within the Methodology section and the Related Works section. This addition includes a comparison with other LLMs, such as GPT and LLaMA, supported by relevant literature. Additionally, the authors have incorporated a comparative analysis in the Discussion section, which highlights the resource trade-offs and specific use-case scenarios where fine-tuning and distillation-based approaches excel. This addition effectively clarifies the merits and limitations of both methods, enhancing the overall clarity and depth of the study.

---

## [Editor Report · Acceptance letter]

16 Jan 2025

PONE-D-24-32612R1 

PLOS ONE

Dear Dr. Karim, 

I'm pleased to inform you that your manuscript has been deemed suitable for publication in PLOS ONE. Congratulations! Your manuscript is now being handed over to our production team.

Kind regards, 

on behalf of

Dr. Weiqiang (Albert) Jin 

Academic Editor

PLOS ONE